# *SlenderGNN*: Accurate, Robust, and Interpretable GNN, and the Reasons for its Success

## Abstract

What is the simplest, but still effective, graph neural network (GNN) that we can design, say, for node classification? Einstein said that we should "make everything as simple as possible, but not simpler." We rephrase it into the *'careful simplicity'* principle: a carefully-designed simple model can outperform sophisticated ones in real-world tasks, where data are scarce, noisy, and spuriously correlated. Based on that principle, we propose *SlenderGNN* that exhibits four desirable properties: It is (a) *accurate*, winning or tying on **11 out of 13** real-world datasets; (b) *robust*, being the **only** one that handles all settings (heterophily, random structure, useless features, etc.); (c) *fast and scalable*, with up to **18×** faster training in million-scale graphs; and (d) *interpretable*, thanks to the linearity and sparsity we impose. We explain the success of *SlenderGNN* via a systematic study on existing models, comprehensive sanity checks, and ablation studies on its design decisions.

← **All reviewers**

## 1 Introduction

What is the simplest, and still performant, graph neural network (GNN) that we can design? GNNs (Kipf & Welling, 2017; Hamilton et al., 2017; Gilmer et al., 2017) have succeeded in various graph mining tasks such as node classification, clustering, or link prediction. However, it is difficult for a practitioner to choose a proper model for each task without spending extensive time on searching, tuning, and training models due to a large number of GNN variants. Given all these variants, which one should a practitioner use first? Which are the strong and weak points of each variant? Could we design a variant that matches all of the strong points and avoids all the weak ones?

In response to the questions above, we propose *SlenderGNN* based on the *'careful simplicity'* principle: a simple, but carefully-designed model can be more accurate than complex ones due to better generalizability, robustness, and easier training. The design decisions of *SlenderGNN* (D1-4 in Section 4.2) are carefully made to follow this principle by observing and addressing the pain points of existing GNNs; for example, we generate various forms of graph-based features and combine them (D1), propose structural features (D2), remove redundancy in the generated features (D3), and make the propagator function contain no hyperparameters (D4). The resulting model, *SlenderGNN*, is our main contribution (C1) which exhibits the following desirable properties:

← **All reviewers**

- **C1.1 - Accurate** on both real-world and synthetic datasets, almost always winning or tying in the first place (see Figure 1b, Table 2, and Table 3).
- **C1.2 - Robust**, being able to handle numerous real settings such as homophily, heterophily, no network effects, graphs with useless features (see Figure 1a and Table 2).
- **C1.3 - Fast and scalable**, using few, carefully chosen features, it takes only 32 seconds on million-scale graphs (ogbn-Products) on a stock server (see Figure 1b).
- **C1.4 - Interpretable**, learning the largest weights on informative features, ignoring noisy ones, based on the linear decision function (see Figure 2).

The natural question that arises from the success of *SlenderGNN* is

← **All reviewers**

> **Q:** "*How is it possible that a simpler model is more accurate than a sophisticated, more expressive one?*"

Our intuitive justification for the success of *SlenderGNN* is as follows: (a) *Occam's razor*: Since a statistical model tries to 'explain' the given labels, the simplest explanation performs best in general.

| Graph | Feature | LR | SGC | DGC | S²GC | G²CN | GCN | SAGE | GCNII | APPNP | GPR | GAT | SlenderGNN |
|---|---|---|---|---|---|---|---|---|---|---|---|---|---|
| Noisy (useless) | Semantic | ✓ | | | ✓ | | | ✓ | | | | | ✓ |
| Homophily | Noisy (useless) | | | | | | | | | | | | ✓ |
| Heterophily | Noisy (useless) | | | | | | | | | | | | ✓ |
| Homophily | Structural | | ✓ | ✓ | ✓ | ✓ | ✓ | ✓ | ✓ | | ✓ | | ✓ |
| Heterophily | Structural | | ✓ | | | ✓ | | ✓ | | | ✓ | | ✓ |
| Homophily | Semantic | ✓ | ✓ | ✓ | ✓ | | ✓ | ✓ | ✓ | ✓ | ✓ | | ✓ |
| Heterophily | Semantic | ✓ | ✓ | | | | ✓ | ✓ | ✓ | | ✓ | | ✓ |

(a) *SlenderGNN* **succeeds in all sanity checks,** while none of the existing models does. The table is generated from the results of our actual experiments in Table 2: ✓ means success (accuracy $\geq 80\%$).

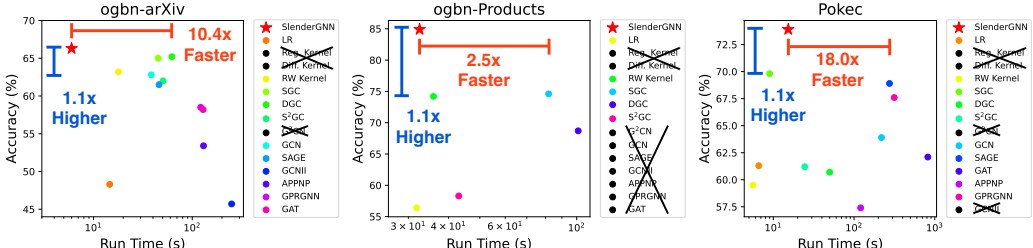

(b) *SlenderGNN* **wins** both on accuracy and training time: it is represented as the red star in (left) ogbn-arXiv, (middle) ogbn-Products and (right) Pokec, which are large real-world graphs (1.2M, 61.9M and 30.6M edges, respectively). Several baselines run out of memory ('crossed out').

Figure 1: *SlenderGNN* **outperforms existing GNN models, is fast, and passes all sanity checks.** See our main results for details (sanity checks in Section 5 and real-world experiments in Section 6).

(b) *Overfitting*: Complex models often suffer from overfitting, exacerbated by the fact that training data are usually scarce and expensive. (c) *Spurious correlations*: Even with sufficient labeled data, a model trained on noisy features may latch on to spurious correlations, which hurt its performance; a simple model effectively suppresses such noisy features as our design decision D3.

In addition to the intuitive arguments above, our extensive experiments provide hard evidence in favor of the *'careful simplicity'* principle: *SlenderGNN* outperforms complex GNNs in both synthetic (in Table 2) and real-world (in Table 3) datasets, and even its own variants that use nonlinear feature transformation in 9 of the 13 real-world datasets (in Table 4).

Not only we propose a carefully designed, high performing GNN, we also explain the reasons of its success. This is thanks to our two additional contributions (C2-3):

- **C2 - Explanation:** We propose GNNLIN, a framework for the systematic linearization of existing GNNs. As shown in Table 1 and Section 3, our GNNLIN highlights the similarities, differences, strengths and weaknesses of successful GNN baselines.
- **C3 - Sanity checks:** We propose a wide range of scenarios (homophily, heterophily, block-communities, bipartite-graph communities, etc.), which reveal the strong and weak points of each GNN variant: see Figure 1a with more details in Table 2 and Section 5.

**Reproducibility:** Our code is available at https://bit.ly/3fhWJfK along with our datasets for 'sanity checks' and real-world datasets of homophily and heterophily graphs.                    ← **R-D5**

## 2   PROBLEM DEFINITION AND RELATED WORKS

We introduce the problem definition of semi-supervised node classification, symbols frequently used in this paper, and related works on graph neural networks (GNN).

**Problem definition**     We define the problem of semi-supervised node classification as follows:

- **Given** An undirected graph $G = (\mathbf{A}, \mathbf{X})$, where $\mathbf{A} \in \mathbb{R}^{n \times n}$ is an adjacency matrix, $\mathbf{X} \in \mathbb{R}^{n \times d}$ is a node feature matrix, $n$ is the number of nodes, and $d$ is the number of features
- **Given** Labels $\mathbf{y} \in \{1, \cdots, c\}^m$ for $m$ nodes, where $m \ll n$, and $c$ is the number of classes.

- **Predict** the unknown classes of $n - m$ test nodes in $G$.

We use the following symbols to represent modified adjacency matrices. $\tilde{\mathbf{A}} = \mathbf{A} + \mathbf{I}$ is the adjacency matrix with self-loops. $\tilde{\mathbf{D}} = \mathrm{diag}(\tilde{\mathbf{A}}\mathbf{1}_{n \times 1})$ is the diagonal degree matrix of $\tilde{\mathbf{A}}$, where $\mathbf{1}_{n \times 1}$ is the matrix of size $n \times 1$ filled with ones. $\tilde{\mathbf{A}}_{\mathrm{sym}} = \tilde{\mathbf{D}}^{-1/2}\tilde{\mathbf{A}}\tilde{\mathbf{D}}^{-1/2}$ is the symmetrically normalized $\tilde{\mathbf{A}}$. Similarly, $\mathbf{A}_{\mathrm{sym}} = \mathbf{D}^{-1/2}\mathbf{A}\mathbf{D}^{-1/2}$ is also the symmetrically normalized $\mathbf{A}$ but without self-loops. There are other types of normalization $\mathbf{A}_{\mathrm{row}} = \mathbf{D}^{-1}\mathbf{A}$ and $\mathbf{A}_{\mathrm{col}} = \mathbf{A}\mathbf{D}^{-1}$ (and accordingly $\tilde{\mathbf{A}}_{\mathrm{row}}$ and $\tilde{\mathbf{A}}_{\mathrm{col}}$), which we call row and column normalization, respectively, based on the position of the degree matrix. Refer to Appendix A for the table of symbols used frequently in this work.

As a background, we formally define logistic regression (LR) as a function to find the weight matrix $\mathbf{W}$ that best maps given features to predicted labels by a linear function.

**Definition 1** (LR). *Given a feature $\mathbf{X} \in \mathbb{R}^{n \times d}$ and a label $\mathbf{y} \in \mathbb{R}^m$, where $m \leq n$ is the number of observations, let $\mathbf{Y} \in \mathbb{R}^{m \times c}$ be the one-hot representation of $\mathbf{y}$, and $y_{ij}$ be the $(i,j)$-th element in $\mathbf{Y}$. Then, logistic regression (LR) finds an optimal weight matrix $\mathbf{W}^* \in \mathbb{R}^{d \times c}$ as follows:*

$$\mathrm{LR}(\mathbf{X}, \mathbf{y}) = \arg\max_{\mathbf{W}} \sum_{i=1}^{l} \sum_{j=1}^{c} y_{ij} \log \hat{y}_{ij} \ \text{ where } \ \hat{y}_{ij} = \frac{\exp(\mathbf{w}_{\cdot j}^{\top}\mathbf{x}_i)}{\sum_{k=1}^{c} \exp(\mathbf{w}_{\cdot k}^{\top}\mathbf{x}_i)}, \tag{1}$$

*and $\mathbf{w}_{\cdot j}$ is the $j$-th column of $\mathbf{W}$. We omit the bias term for brevity without loss of generality.*

## 2.1 RELATED WORKS

*Graph neural networks*  There are many recent GNN variants; recent surveys (Zhou et al., 2020; Wu et al., 2021) group them into spectral models (Defferrard et al., 2016; Kipf & Welling, 2017), sampling-based models (Hamilton et al., 2017; Ying et al., 2018), attention-based models (Velickovic et al., 2018; Kim & Oh, 2021; Brody et al., 2022), and deep models with residual connections (Li et al., 2019; Chen et al., 2020). Decoupled models (Klicpera et al., 2019a;b; Chien et al., 2021) separate the two major functionalities of GNNs: the node-wise feature transformation and the propagation. GNNs are often fused with graphical inference (Yoo et al., 2019; Huang et al., 2021).

*Linear graph neural networks*  Wu et al. (2019) proposed SGC by removing the nonlinear activation functions of GCN (Kipf & Welling, 2017) reducing the propagator function to a simple matrix multiplication. Wang et al. (2021) and Zhu & Koniusz (2021) improved SGC by manually adjusting the strength of self-loops with hyperparameters, increasing the number of propagation steps. Li et al. (2022) proposed G$^2$CN, which improves the performance of DGC (Wang et al., 2021) on heterophily graphs by combining multiple propagation settings (i.e. bandwidths). The main limitation of these models is the high complexity of propagator functions with many hyperparameters, which impairs both the robustness and interpretability of decisions even with linearity.

*Graph kernel methods*  Traditional works on graph kernel methods (Smola & Kondor, 2003; Ioannidis et al., 2017) are closely related to linear GNNs, which can be understood as applying a linear graph kernel to transform the raw features. A notable limitation of such kernel methods is that they are not capable of addressing various scenarios of real-world graphs, such as heterophily graphs, as their motivation is to aggregate all information in the local neighborhood of each node, rather than ignoring noisy and useless ones. We implement three popular kernel methods as additional baselines and show that our *SlenderGNN* outperforms them in both synthetic and real graphs.

$\leftarrow$ **R-V2**

## 3 PROPOSED FRAMEWORK: GNNLIN

Why do GNNs work well when they do? In what cases will a GNN fail? We answer these questions with our proposed GNNLIN, which reveals the essence of each GNN variant. The idea is to derive the feature propagator function that each variant uses ignoring nonlinearity. Our observations help us do the careful design of our *SlenderGNN*, which we describe in Section 4.

**Definition 2** (Linearized GNN). *Given a graph $G = (\mathbf{A}, \mathbf{X})$, let $f(\cdot; \theta)$ be a node classifier function to predict the labels of all nodes in $G$ as $\hat{\mathbf{y}} = f(\mathbf{A}, \mathbf{X}; \theta)$, where $\theta$ is the set of learnable parameters. Then, $f$ is linearized if $\theta = \{\mathbf{W}\}$ and an optimal weight matrix $\mathbf{W}^* \in \mathbb{R}^{h \times c}$ is given as*

$$\mathbf{W}^* = \mathrm{LR}(\mathcal{P}(\mathbf{A}, \mathbf{X}), \mathbf{y}), \tag{2}$$

Table 1: **GNNLIN framework is general** encompassing popular GNN models. The * and ** superscripts mark fully and partially linearized models, respectively; see Section 3.1 for details.

| Model | Type | Propagator function $\mathcal{P}(\mathbf{A}, \mathbf{X})$ | Hyperparameters |
|---|---|---|---|
| LR | Linear | $\mathbf{X}$ | - |
| SGC | Linear | $\tilde{\mathbf{A}}_{\text{sym}}^K \mathbf{X}$ | $K$ |
| DGC | Linear | $[(1 - T/K)\mathbf{I} + (T/K)\tilde{\mathbf{A}}_{\text{sym}}]^K \mathbf{X}$ | $K, T$ |
| S²GC | Linear | $\sum_{k=1}^{K} (\alpha\mathbf{I} + (1-\alpha)\tilde{\mathbf{A}}_{\text{sym}}^k)\mathbf{X}$ | $K, \alpha$ |
| G²CN | Linear | $\|_{i=1}^{N}[\mathbf{I} - (T_i/K)((b_i - 1)\mathbf{I} + \mathbf{A}_{\text{sym}})^2]^K \mathbf{X}$ | $K, N, T_i, b_i$ |
| PPNP* | Decoupled | $(\mathbf{I} - (1-\alpha)\tilde{\mathbf{A}}_{\text{sym}})^{-1}\mathbf{X}$ | $\alpha$ |
| APPNP* | Decoupled | $[\sum_{k=0}^{K-1} \alpha(1-\alpha)^k \tilde{\mathbf{A}}_{\text{sym}}^k + (1-\alpha)^K \tilde{\mathbf{A}}_{\text{sym}}^K]\mathbf{X}$ | $K, \alpha$ |
| GDC* | Decoupled | $\tilde{\mathbf{S}}_{\text{sym}}\mathbf{X}$ where $\mathbf{S} = \text{sparse}_\epsilon(\sum_{k=0}^{\infty}(1-\alpha)^k \tilde{\mathbf{A}}_{\text{sym}}^k)$ | $\alpha, \epsilon$ |
| GPR-GNN* | Decoupled | $\|_{k=0}^{K} \tilde{\mathbf{A}}_{\text{sym}}^k \mathbf{X}$ | $K$ |
| ChebNet* | Coupled | $\|_{k=0}^{K-1} \mathbf{A}_{\text{sym}}^k \mathbf{X}$ | $K$ |
| GCN* | Coupled | Same as SGC | $K$ |
| SAGE* | Coupled | $\|_{k=0}^{K} \mathbf{A}_{\text{row}}^k \mathbf{X}$ | $K$ |
| GCNII* | Coupled | $\|_{k=0}^{K-2} \tilde{\mathbf{A}}_{\text{sym}}^k \mathbf{X} \| ((1-\alpha)\tilde{\mathbf{A}}_{\text{sym}}^K + \alpha\tilde{\mathbf{A}}_{\text{sym}}^{K-1})\mathbf{X}$ | $K, \alpha$ |
| GAT** | Attention | $\prod_{k=1}^{K}[\text{diag}(\mathbf{X}\mathbf{w}_{k,1})\tilde{\mathbf{A}} + \tilde{\mathbf{A}}\text{diag}(\mathbf{X}\mathbf{w}_{k,2})]\mathbf{X}$ | $K, \mathbf{w}_{k,1}, \mathbf{w}_{k,2}$ |
| DA-GNN** | Attention | $\sum_{k=0}^{K} \text{diag}(\tilde{\mathbf{A}}_{\text{sym}}^k \mathbf{X}\mathbf{w})\tilde{\mathbf{A}}_{\text{sym}}^k \mathbf{X}$ | $K, \mathbf{w}$ |

*where $\mathcal{P}$ is a feature propagator function that is linear with $\mathbf{X}$ and contains no learnable parameters, and $\mathcal{P}(\mathbf{A}, \mathbf{X}) \in \mathbb{R}^{n \times h}$. We ignore the bias term for brevity without loss of generality.*

**Definition 3** (GNNLIN). *Let $f(\cdot; \theta)$ be a (nonlinear) GNN. GNNLIN is to represent $f$ as a linearized GNN by replacing all (nonlinear) activation functions in $f$ with the identity function and deriving a variant $f'$ that is at least as expressive as $f$ but contains no parameters in $\mathcal{P}$.*

GNNLIN represents the characteristic of a GNN as the linear feature propagation function $\mathcal{P}$, which transforms raw features $\mathbf{X}$ by utilizing $\mathbf{A}$. Lemma 1 shows that GNNLIN generalizes existing linear GNNs. Logistic regression is also represented by GNNLIN with the identity $\mathcal{P}(\mathbf{A}, \mathbf{X}) = \mathbf{X}$.

**Lemma 1.** *Our GNNLIN framework includes existing linear GNN models as its special cases: SGC, DGC, S²GC, and G²CN.*

*Proof.* The proof is given in Appendix B. ∎

### 3.1 LESSONS FROM GNNLIN

Table 1 shows the linearized form of existing GNNs generated from our GNNLIN framework. Refer to Appendix C for the detailed information. Based on the result, we spot the fundamental similarities and differences among the GNN variants. There are three distinguishing factors.

**Distinguishing Factor 1** (Combination of features). *How should we combine the node features, the immediate neighbors' features, and the $K$-step-away neighbors' features?*

GNNs propagate information by multiplying the feature $\mathbf{X}$ with (a variant of) the adjacency matrix $\mathbf{A}$ multiple times. There are two main choices in Table 1: **(1)** summation of the transformed features up to $K$ steps (most models), and **(2)** concatenation (GPR-GNN, GraphSAGE, and GCNII). Simple approaches like SGC are categorized as summation due to the self-loops in $\tilde{\mathbf{A}}_{\text{sym}}$.

**Distinguishing Factor 2** (Modification of adjacency matrices). *How should we normalize or modify the adjacency matrix?*

The three prevailing choices are **(1)** symmetric vs. row normalization, **(2)** the strength of self-loops, including making zero self-loops, and **(3)** static vs. dynamic adjustment based on the given features. Most models use the symmetric normalization $\tilde{\mathbf{A}}_{\text{sym}}$ with self-loops, but some variants avoid self-loops and use either row normalization $\mathbf{A}_{\text{row}}$ or symmetric one $\mathbf{A}_{\text{sym}}$. Recent models such as DGC, G²CN, and GCNII determine the weight of self-loops with hyperparameters, since strong self-loops

allow one to increase the value of $K$ for distant propagation. Finally, attention-based models learn the elements in $\mathbf{A}$ based on node features, making propagator functions quadratic with $\mathbf{X}$.

**Distinguishing Factor 3** (Heterophily). *What to do if the direct neighbors differ in their features or labels?*

In such cases, the simple aggregation of the features of immediate neighbors may hurt performance, and therefore, several GNNs do suffer under heterophily as shown in Figure 1a and Table 2. GNNs that can handle heterophily adopt one or more of these ideas: **(1)** using the square of $\mathbf{A}$ as the base structure (in $G^2CN$; "the enemies of my enemy are my friends"); **(2)** learning different weights for different steps (GPR-GNN, ChebNet, SAGE, and GCNII), and **(3)** making small or no self-loops in the $\mathbf{A}$ matrix (DGC, $S^2GC$, and $G^2CN$). The idea is to avoid or downplay the effect of immediate (and odd-step-away) neighbors. Self-loops hurt under heterophily, as they force to have information of all intermediate neighbors by acting as implicit summation of transformed features.

## 4 PROPOSED METHOD: *SlenderGNN*

We propose *SlenderGNN*, a novel GNN model that addresses the limitations of existing GNNs with the strict adherence to the *'careful simplicity'* principle. We first present the pain points of existing GNNs derived from Table 1 and then describe how *SlenderGNN* addresses them.

### 4.1 PAIN POINTS OF EXISTING GNNS

**Pain Point 1** (Lack of robustness). *All models in Table 1 fail to handle multiple graph scenarios at the same time, i.e., graphs with homophily, heterophily, no network effects, or useless features.*    ← **R-f2, D3**

Models in Table 1 assume a specific scenario, such as homophily or heterophily graphs, rather than being able to perform in multiple scenarios at the same time. For example, all of these models except ChebNet and SAGE include the self-loops in the updated adjacency matrix, emphasizing the local neighborhood even in graphs with heterophily or no network effects. This is the pain point that we also observe empirically from the sanity checks (in Figure 1a and Table 2).

**Pain Point 2** (Failure on noisy features). *All models in Table 1 depend on the node feature matrix $\mathbf{X}$, and cannot fully exploit the adjacency matrix $\mathbf{A}$ if the given features are noisy.*    ← **R-f2, D3**

Real-world datasets often contain noisy features, and the graph structure $\mathbf{A}$ plays an essential role in the performance of node classification in such cases. That is, a desirable property for a robust model is to adaptively emphasize important features or disregard noisy ones to maximize its generalization performance. However, the models in Table 1 lack such a functionality.

**Pain Point 3** (Hyperparameters in propagators). *The hyperparameters in a propagator function $\mathcal{P}$ impair the interpretability of the weight matrix $\mathbf{W}$, and force the re-computation of the transformed feature for every new choice during hyperparameter search.*

The last column of Table 1 summarizes the hyperparameters in each $\mathcal{P}$, which make the following limitations in terms of linear models. First, the interpretability of the weight matrix $\mathbf{W}$ is impaired, because it is learned on the transformed feature $\mathcal{P}(\mathbf{A}, \mathbf{X})$ whose meaning changes arbitrarily by the choice of hyperparameters. Second, $\mathcal{P}(\mathbf{A}, \mathbf{X})$ should be computed for each choice of hyperparameters, while it can be cached and reused for searching hyperparameters outside $\mathcal{P}$.

### 4.2 DESIGN DECISIONS OF *SlenderGNN*

**Summary:**    Considering all the pain points and adhering to the *'careful simplicity'* principle, we make *design decisions* (D1-D4) that lead to the following propagator function $\mathcal{P}$ of *SlenderGNN*:

$$\mathcal{P}(\mathbf{A}, \mathbf{X}) = \underbrace{\mathbf{U}}_{\text{Structure}} \parallel \underbrace{g(\mathbf{X})}_{\text{Node features}} \parallel \underbrace{g(\mathbf{A}^2_{\text{row}}\mathbf{X})}_{\text{2-step neighbors}} \parallel \underbrace{g(\tilde{\mathbf{A}}^2_{\text{sym}}\mathbf{X})}_{\text{Neighbors}} \tag{3}$$

where $g(\cdot)$ is the principal component analysis (PCA) for the orthogonalization of each component, followed by an L2 normalization, and $\mathbf{U} \in \mathbb{R}^{n \times r}$ contains $r$-dimensional structural features derived by running the low-rank singular value decomposition (SVD) on the adjacency matrix $\mathbf{A}$.

**D1**: Concatenation of winning normalizations    The main principle of *SlenderGNN* to acquire the robustness and generalizability, in response to Pain Point 1, is to transform the raw features into various forms and then combine them by concatenation. In this way, *SlenderGNN* is able to emphasize essential features or ignore useless ones by learning separate weights for different components. The four components of Equation 3 are proposed to have their strength in different scenarios: the structural information $\mathbf{U}$, self-feature information $\mathbf{X}$, two-step aggregation $\mathbf{A}_{\mathrm{row}}^2$ for heterophily graphs, and the smoothed two-hop aggregation $\tilde{\mathbf{A}}_{\mathrm{sym}}^2$ of local neighborhood, respectively.    ← **R-f2, D3**

Specifically, we use the row-normalized matrix $\mathbf{A}_{\mathrm{row}}$ with no self-loops due to the limitations of the symmetric normalization $\tilde{\mathbf{A}}_{\mathrm{sym}}$: First, the self-loops force one to combine all intermediate neighbors until the $K$-hop distance, even in heterophily graphs where the direct neighbors should be avoided. Second, the neighboring features are rescaled based on the node degrees during an aggregation, even when we want simple aggregation of $K$-hop neighbors preserving the original scale.

**D2**: Structural features    In response to Pain Point 2 where features are missing, noisy, or useless for classification, then we have to resort to the adjacency matrix $\mathbf{A}$ ignoring $\mathbf{X}$. At the same time, it is not effective to use raw $\mathbf{A}$, which is a large sparse matrix. We thus adopt low-rank SVD with rank $r$ to extract structural features $\mathbf{U}$. The value of $r$ is automatically selected to keep $90\%$ of the energy of $\mathbf{A}$, where the sum of the largest $r$ squared singular values divided by the squared Frobenius norm of $\mathbf{A}$ is approximately $0.9$. When the graph is large, we set $r$ to be $d$ for the size consistency.    ← **R-f2, D3**

**D3**: Orthogonalization and sparsification    We use two reliable methods to further address Pain Point 2 about noisy features: dimensionality reduction by PCA and regularization by group LASSO. First, we run PCA on each component independently to orthogonalize given features and to improve the consistency of learned weights. Second, we apply group LASSO to learn sparse weights on the component level, preserving the relative magnitude of each element and suppressing noisy features. To make the consistency between multiple components, we force all components to have the same dimensionality by selecting $r$ features from each component when adopting PCA.    ← **R-f2, D3**

**D4**: No hyperparameters in the propagator $\mathcal{P}$    We address Pain Point 3 by making the propagator function $\mathcal{P}$ contain no hyperparameters to tune for each dataset. Most models in Table 1 contain such hyperparameters. For example, DGC is a linear model, but its interpretability is limited since it selects $K$ and $T$ from arbitrary values, i.e., $K \in \{250, 300, 900\}$ and $T \in \{5.27, 3.78, 6.0498\}$. On the contrary, our effective design of $\mathcal{P}$ (i.e., D1 and D2) allows us to keep the small value of $K = 2$ in all 13 datasets we use in the experiments without sacrificing its performance.

## 5   PROPOSED SANITY CHECKS

We propose *sanity checks* to evaluate the robustness of GNNs to various scenarios of node classification and to observe their strengths and weaknesses in different settings.

**Graph scenarios**    We categorize possible scenarios of node classification based on the characteristics of node features $\mathbf{X}$, a graph structure $\mathbf{A}$, and node labels $\mathbf{y}$. We describe only the main ideas, leaving exact definitions of such scenarios to Appendix D.

- **Features X:** We consider three cases: *random*, *structural*, and *semantic*. The random case means that each feature is determined independently of all other variables. In the structural case, features give information of the graph structure, and in the semantic case, they directly provide useful information for node labels. Unlike the cases of edges and labels, which are mutually exclusive, features can be both structural and semantic at the same time.
- **Edges A:** We consider three cases: *uniform* (no communities), *clustered* (block-diagonal), and *bipartite*. The uniform case means that every element $a_{ij}$ is determined independently of the other edges in $\mathbf{A}$. In the clustered case, nodes having common neighbors are likely to make more edges, while it is the opposite in the bipartite case.
- **Labels y:** We consider three cases: *individual* (no network effects, i.e., there is no predictive power of connectivity), *homophily*, and *heterophily*. In the homophily case, adjacent nodes are likely to have the same label, while it is the opposite in the heterophily case.

**Feasibility**    Although there exist a total of 27 combinations for $(\mathbf{X}, \mathbf{A}, \mathbf{y})$, not all of them are possible to implement; for example, either homophily or heterophily $\mathbf{y}$ is not compatible with uniform

Table 2: *SlenderGNN* **passes all sanity checks.** The accuracy of all models on sanity checks; there are three groups of scenarios: (left) only features **X** help; (middle) only connectivity **A** helps; (right) both help. See the text for details on S, U, I, C, B, etc. Green (■, ■, ■) marks the top three (higher is darker); red (■) marks the ones that are too low ($2\sigma$ below the third place).

| Model | X helps and A is useless | | | A helps and X is useless | | Both X and A help | | | |
|---|---|---|---|---|---|---|---|---|---|
| | (S, U, I) Individual | (S, C, I) Individual | (S, B, I) Individual | (R, C, O) Homophily | (R, B, E) Heterophily | (T, C, O) Homophily | (T, B, E) Heterophily | (S, C, O) Homophily | (S, B, E) Heterophily |
| LR | 83.7±0.6 | 83.7±0.6 | 83.7±0.6 | 24.2±0.7 | 24.2±0.7 | 71.4±0.9 | 66.8±2.2 | 83.4±0.6 | 83.4±0.6 |
| Reg. Kernel | 82.7±0.5 | 82.3±0.6 | 82.7±0.4 | 27.9±0.4 | 24.3±1.0 | 75.7±0.2 | 65.3±1.6 | 91.5±0.5 | 79.5±0.3 |
| Diff. Kernel | 26.8±1.7 | 27.6±2.3 | 27.3±1.8 | 38.0±8.7 | 37.6±7.5 | 79.5±0.3 | 73.5±0.6 | 70.9±23. | 56.1±27. |
| RW Kernel | 72.2±0.7 | 72.2±1.9 | 73.0±0.8 | 37.0±0.4 | 24.5±1.3 | 81.3±1.2 | 51.0±1.1 | 94.5±0.9 | 57.8±0.7 |
| SGC | 44.6±9.8 | 43.0±9.0 | 45.1±10. | 64.3±0.7 | 50.2±14. | 87.1±0.6 | 84.3±0.5 | 93.9±0.9 | 91.5±0.5 |
| DGC | 63.8±1.0 | 66.3±0.9 | 65.9±1.2 | 50.5±13. | 26.0±0.9 | 88.6±1.0 | 45.3±1.3 | 96.2±0.4 | 54.0±0.6 |
| S²GC | 79.9±0.6 | 79.8±1.1 | 80.1±1.0 | 38.5±12. | 25.4±0.9 | 88.4±1.0 | 67.9±1.5 | 95.9±0.6 | 78.0±0.5 |
| G²CN | 25.2±0.3 | 25.3±0.1 | 25.0±0.2 | 24.2±1.1 | 25.0±0.1 | 88.5±1.0 | 88.6±1.2 | 24.3±1.1 | 50.7±31. |
| GCN | 36.3±3.5 | 33.2±2.4 | 35.7±3.5 | 46.7±8.0 | 43.7±1.9 | 83.3±1.3 | 72.2±1.7 | 91.2±1.2 | 80.3±3.9 |
| SAGE | 80.3±1.1 | 79.0±1.2 | 79.8±0.7 | 31.1±0.7 | 34.6±2.1 | 83.9±0.8 | 81.3±0.7 | 94.4±0.5 | 94.4±0.9 |
| GCNII | 73.5±1.2 | 73.2±0.9 | 73.6±0.9 | 30.7±0.7 | 27.1±1.3 | 84.2±0.8 | 69.0±1.4 | 90.6±0.9 | 80.4±1.2 |
| APPNP | 66.0±2.6 | 65.4±2.7 | 64.2±1.6 | 30.3±1.2 | 25.2±0.7 | 71.2±4.9 | 43.8±2.0 | 83.2±3.8 | 58.7±4.5 |
| GPR-GNN | 73.4±0.4 | 73.5±1.3 | 73.9±0.7 | 74.6±0.7 | 65.9±2.1 | 89.9±0.6 | 87.6±1.2 | 95.0±1.1 | 91.9±1.1 |
| GAT | 32.7±5.5 | 30.6±3.0 | 33.8±6.8 | 42.6±4.8 | 36.8±5.7 | 64.0±5.7 | 55.6±6.8 | 68.5±7.1 | 67.0±12. |
| *SlenderGNN* | 81.0±1.1 | 81.0±0.9 | 81.2±1.0 | 87.1±1.4 | 89.2±1.2 | 88.1±0.5 | 88.9±0.7 | 94.4±0.6 | 93.9±0.5 |

**A.** After removing the infeasible combinations of variables, we categorize the remaining choices based on the predictive power of structure (**A**) and features (**X**) for labels (**y**).

**Results of sanity checks** Table 2 shows the results of sanity checks for our *SlenderGNN* and all baseline models (details of the baselines in Section 6). ■, ■, ■ represent top three methods (higher is darker) with overlap within $2\sigma$, and ■ represents accuracy below $2\sigma$ of the third-best method. We run each experiment five times and report the average and standard deviation. We assume four target classes of nodes, and thus the accuracy of random guessing is 25%. The letters represent the cases of **X**, **A**, and **y**, respectively: S (semantic **X**), R (random **X**), T (structural **X**), U (uniform **A**), C (clustered **A**), B (bipartite **A**), I (individual **y**), O (homophily **y**), and E (heterophily **y**).

It is clear that our *SlenderGNN* is the only approach that passes all sanity checks. The four components in *SlenderGNN* are carefully designed to maximize its robustness for various graph scenarios. Most GNNs work well in the cases (S, C, O) and (T, C, O), where **X** is either semantic or structural, **A** is clustered, and **y** is homophily, since many real-world datasets that recent works on GNNs use in their experiments follow such assumptions. However, many GNNs fail when a graph is generated with different assumptions, as we summarize as follows:

- **No network effects:** In the cases of (S, ?, I), where ? is a placeholder, only a few models such as SAGE and GPR-GNN perform well. This is because **A** is not informative in such cases, and models are required to focus on raw features **X** ignoring **A**.
- **Useless features:** In the cases of (R, C, O) and (R, B, E), models are required to do the opposite since **X** is not informative: they should focus on **A**, ignoring **X**. Since there is no approach that explicitly uses **A**, all baselines show low accuracy.
- **Heterophily graphs:** In the cases of (?, B, E), where ? is a placeholder, the labels follow heterophily. Models like G²CN, SAGE, and GPR-GNN perform well in these cases, since they address the heterophily in their designs (details in Section 3.1).

## 6 EXPERIMENTS

We perform experiments on 13 real-world datasets to answer the following research questions (RQ):

RQ1. **Accuracy:** How well does *SlenderGNN* work on real-world graphs?
RQ2. **Speed and Scalability:** How fast and scalable is *SlenderGNN* to large real-world graphs?
RQ3. **Interpretability:** How to interpret the learned weights of *SlenderGNN*?
RQ4. **Ablation Study:** Are all the design decisions of *SlenderGNN* effective in real-world graphs? What if we add nonlinearity to *SlenderGNN* or increase its receptive field?

Table 3: *SlenderGNN* **wins** most of the times on 13 real-world datasets (7 homophily and 6 heterophily graphs) against 14 competitors. We color the best and worst results as in Table 2.

| Model | Cora | CiteSeer | PubMed | Comp. | Photo | ArXiv | Products | Cham. | Squirrel | Actor | Penn94 | Twitch | Pokec |
|---|---|---|---|---|---|---|---|---|---|---|---|---|---|
| LR | 51.5±1.2 | 52.9±4.5 | 79.9±0.5 | 73.9±1.2 | 79.3±1.5 | 48.3±1.9 | 56.4±0.5 | 24.9±1.7 | 26.7±1.9 | 27.8±0.8 | 63.5±0.5 | 53.0±0.1 | 61.3±0.0 |
| Reg. Kernel | 67.8±2.5 | 62.1±4.4 | 83.4±1.4 | 80.3±1.4 | 87.1±1.2 | O.O.M. | O.O.M. | 29.4±2.6 | 24.3±2.3 | 29.6±1.4 | O.O.M. | O.O.M. | O.O.M. |
| Diff. Kernel | 70.6±1.5 | 62.7±3.8 | 82.1±0.4 | 83.1±1.0 | 89.8±0.6 | O.O.M. | O.O.M. | 34.5±7.9 | 28.3±1.5 | 24.7±0.9 | 53.5±0.8 | O.O.M. | O.O.M. |
| RW Kernel | 72.7±1.7 | 64.1±3.9 | 83.1±0.7 | 84.2±0.7 | 90.6±0.7 | 63.2±0.2 | 74.2±0.0 | 34.9±3.5 | 25.0±1.6 | 26.4±1.1 | 63.1±0.7 | 57.6±0.1 | 59.5±0.0 |
| SGC | 76.2±1.1 | 65.8±3.9 | 84.1±0.8 | 83.7±1.6 | 90.1±0.9 | 65.0±3.4 | 74.6±5.1 | 38.1±4.5 | 33.1±1.0 | 24.6±0.8 | 64.0±1.1 | 56.5±0.1 | 69.8±0.0 |
| DGC | 77.8±1.4 | 66.1±4.2 | 84.3±0.6 | 83.9±0.7 | 90.4±0.2 | 65.2±4.0 | 68.7±13. | 37.2±3.7 | 29.2±1.2 | 25.2±2.1 | 62.5±0.4 | 58.2±0.2 | 60.7±0.1 |
| S²GC | 78.3±1.5 | 66.9±4.4 | 84.3±0.3 | 83.1±0.8 | 90.1±0.8 | 62.0±7.4 | 58.3±18. | 34.9±4.9 | 27.6±1.8 | 26.7±1.8 | 63.1±0.5 | 58.7±0.1 | 61.2±0.0 |
| G²CN | 76.6±1.5 | 64.2±3.3 | 81.4±0.6 | 82.8±1.6 | 88.8±0.5 | O.O.M. | O.O.M. | 40.7±2.9 | 32.1±1.5 | 24.3±0.5 | O.O.M. | O.O.M. | O.O.M. |
| GCN | 76.0±1.2 | 65.0±2.9 | 84.3±0.5 | 85.1±0.9 | 91.6±0.5 | 62.8±0.6 | O.O.M. | 38.5±3.0 | 31.4±1.8 | 26.8±0.4 | 62.9±0.7 | 57.0±0.1 | 63.9±0.4 |
| SAGE | 74.6±1.3 | 63.7±3.6 | 82.9±0.4 | 83.8±0.5 | 90.6±0.5 | 61.5±0.6 | O.O.M. | 39.8±4.3 | 27.0±1.3 | 27.8±0.9 | O.O.M. | 56.6±0.4 | 68.9±0.1 |
| GCNII | 77.8±1.7 | 63.4±3.0 | 84.9±0.8 | 82.3±1.8 | 90.8±0.6 | 45.7±0.5 | O.O.M. | 30.5±2.5 | 21.9±3.0 | 29.0±1.3 | 64.5±0.5 | 56.9±0.6 | 62.1±0.3 |
| APPNP | 80.0±0.6 | 67.1±2.8 | 84.6±0.5 | 84.2±1.7 | 92.5±0.3 | 53.4±1.3 | O.O.M. | 30.9±4.7 | 23.9±3.2 | 26.1±1.0 | 63.7±0.9 | 47.3±0.3 | 57.4±0.4 |
| GPR-GNN | 78.8±1.3 | 64.2±4.0 | 85.1±0.7 | 85.0±1.0 | 92.6±0.3 | 58.5±0.8 | O.O.M. | 31.7±4.7 | 26.2±1.6 | 29.5±1.1 | 64.5±0.4 | 57.6±0.2 | 67.6±0.1 |
| GAT | 78.2±1.2 | 65.8±4.0 | 83.6±0.2 | 85.4±1.4 | 91.7±0.5 | 58.2±1.0 | O.O.M. | 39.1±4.1 | 28.6±0.6 | 26.4±0.4 | 60.5±0.8 | O.O.M. | O.O.M. |
| *SlenderGNN* | 77.8±1.1 | 67.1±2.3 | 84.6±0.5 | 86.3±0.7 | 91.8±0.5 | 66.3±0.3 | 84.9±0.0 | 40.8±3.2 | 31.1±0.7 | 30.9±0.6 | 68.2±0.6 | 59.7±0.1 | 73.9±0.1 |

**Datasets and competitors**   We use 7 homophily and 6 heterophily graphs in experiments, which were commonly used in previous works on node classification (Chien et al., 2021; Pei et al., 2020; Lim et al., 2021). We adopt various types of models as competitors: linear GNNs (LR, SGC, DGC, S²GC, and G²CN), coupled nonlinear models (GCN, GraphSAGE, and GCNII), decoupled models (APPNP and GPR-GNN), and attention-based models (GAT). We also include three graph kernel   ← **R-V2** methods (Smola & Kondor, 2003), namely Regularized Laplacian, Diffusion Process, and the $K$-step Random Walk. They are used with LR as *SlenderGNN* is. We perform hyperparameter search based on those reported in their original papers. Refer to Appendix E for details.

**Experimental setup**   We perform semi-supervised node classification by dividing all nodes in a graph by the $2.5\%/2.5\%/95\%$ ratio into training, validation, and testing data. We perform five runs of each experiment with different random seeds and report the average and standard deviation. All hyperparameter search and early stopping are done based on validation accuracy for each run.

**RQ1. Accuracy**   In Table 3, *SlenderGNN* is compared against linear as well as the state-of-the-art nonlinear GNNs on 13 real-world datasets (7 homophily and 6 heterophily graphs). We report the accuracy in Table 3 where ■, ■, ■ represent the top three methods (higher is darker), ■ represents the accuracy below $2\sigma$ of the third-best method, and ■ represents the out-of-memory error (O.O.M.). Our *SlenderGNN* outperforms all competitors in **4** homophily and **5** heterophily datasets, and shows competitive accuracy in the rest (11 out of 13 times among top three methods). Moreover, *SlenderGNN* is the only model without red cells, which demonstrates its robustness and generality. Many competitors run out of memory when the graph reaches the million-edge scale.

**RQ2. Speed and scalability**   We plot the training time versus the accuracy of each model on the   ← **R-V3, y3** ogbn-arXiv, ogbn-Products, and Pokec datasets, which are largest in our benchmark, in Figure 1b. We report the training time of each model with the hyperparameters that show the highest validation accuracy. *SlenderGNN* achieves the highest accuracy in the ogbn-arXiv and ogbn-Products datasets, while being $10.4\times$ and $2.5\times$ faster than the second-best model, respectively. *SlenderGNN* also shows the highest accuracy in the Pokec dataset, which is a large heterophily graph, while being $18.0\times$ faster than the best performing deep model. It is worth noting that *SlenderGNN* is even faster than LR in ogbn-arXiv, requiring only half the number of iterations that LR needs while optimizing. Its fast convergence is owing to orthogonalization on each component of the features.

**RQ3. Interpretability**   Figure 2 illustrates the learned weights of our *SlenderGNN* for the sanity   ← **R-V3, D4** checks, where the ground truths are known. *SlenderGNN* assigns large weights to the correct factors in graphs with different mutual information between variables. When there are no network effects in Figure 2a, it successfully assigns the largest weights to the self-features $g(\mathbf{X})$, ignoring all other components. When the features are useless in Figure 2b, it puts most of the attention on the structural features $\mathbf{U}$. In the left two of Figure 2c, when the features are useful but largely correlated with the structure, it ignores the self-features $g(\mathbf{X})$ and assigns the largest weights to the graph-propagated features. In the right two of Figure 2c, when every component is informative, *SlenderGNN* assigns large weights to both self- and propagated features to maximize its accuracy.

**RQ4. Ablation studies**   We perform three ablation studies to better understand how *SlenderGNN*   ← **R-V1, V3** works: its (a) linearity, (b) receptive field, and (c) four different components. We show the result on

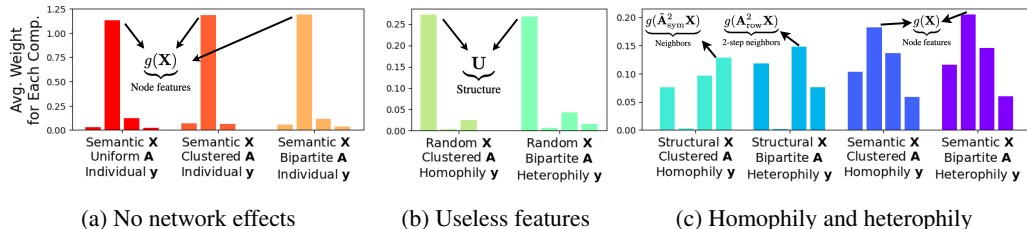

(a) No network effects     (b) Useless features     (c) Homophily and heterophily

Figure 2: *SlenderGNN* **is interpretable**: it suppresses useless information and focuses on the informative ones for each scenario: (a) self-features, (b) structural features, and (c) mixed.

Table 4: **Ablation Study - *SlenderGNN* works best with linearity:** *SlenderGNN* outperforms its own variants that replace the linear classifier or the PCA function $g$ with a nonlinear module.

| Model | Cora | CiteSeer | PubMed | Comp. | Photo | ArXiv | Products | Cham. | Squirrel | Actor | Penn94 | Twitch | Pokec |
|---|---|---|---|---|---|---|---|---|---|---|---|---|---|
| w/ MLP-2 | 65.9±1.1 | 54.2±5.3 | 83.3±0.2 | 84.8±0.5 | 90.1±1.9 | 65.8±0.1 | **85.7±0.0** | 40.0±2.5 | 30.5±0.5 | 28.8±1.0 | 66.7±1.7 | 60.3±0.3 | **76.5±0.1** |
| w/ MLP-3 | 66.1±2.0 | 50.3±3.6 | 80.9±0.9 | 85.2±0.8 | 90.0±0.8 | 63.0±0.1 | 84.9±0.9 | 38.5±5.5 | 30.9±0.7 | 28.9±1.2 | 65.1±0.6 | 60.3±0.2 | 76.4±0.2 |
| w/ NL Trans. | 70.7±2.3 | 57.5±5.1 | 81.0±0.4 | 71.4±10. | 77.9±2.2 | 57.0±0.6 | O.O.M. | **41.3±3.2** | 30.0±1.6 | 27.6±2.4 | 61.8±1.6 | **61.5±0.3** | 75.7±0.5 |
| *SlenderGNN* | **77.8±1.1** | **67.1±2.3** | **84.6±0.5** | **86.3±0.7** | **91.8±0.5** | **66.3±0.3** | 84.9±0.0 | 40.8±3.2 | **31.1±0.7** | **30.9±0.6** | **68.2±0.6** | 59.7±0.1 | 73.9±0.1 |

linearity in Table 4, and give the results of the last two experiments in Table 8 and 9, respectively, in Appendix F. Detailed information on the settings of ablation studies is also given in Appendix F. In short, we observe that (a) the linear version performs best, (b) the two-step aggregation is sufficient, and (c) all four components in *SlenderGNN* are essential in its performance.

Specifically, the success of *SlenderGNN* against its nonlinear variants in Table 4 supports the power of the *'careful simplicity'* principle. The increased expressiveness harms the accuracy since (a) the original *SlenderGNN* is already effective to capture necessary information for classification, and (b) an increased number of parameters cause the loss of generality through overfitting. Needless to say, the nonlinear variants take longer time in training, have more hyperparameters to tune, and lose the interpretability which is the great advantage of linear models (as we show in Figure 2).

# 7 CONCLUSION

The main contribution (C1) of this work is *SlenderGNN*, which is designed by the *'careful simplicity'* principle, and thus has a long list of desirable properties:

- **C1.1 - Accurate:** On both synthetic and real graphs, *SlenderGNN* exceeds the accuracy of state-of-the-art linear GNNs and matches the accuracy of nonlinear models.
- **C1.2 - Robust:** *SlenderGNN* succeeds in graphs with homophily, heterophily, no network effects (i.e., random connections), and no meaningful features.
- **C1.3 - Fast and scalable:** *SlenderGNN* is scalable to million-scale graphs, where most of the existing models run out of memory, with up to $18\times$ less training time.
- **C1.4 - Interpretable:** *SlenderGNN* automatically selects important features and can justify its decisions based on the learned weights, thanks to its linearity.

Additional contributions focus on explaining the success of *SlenderGNN*:

- **C2 - Explanation:** The GNNLIN framework illuminates the fundamental similarities and differences of popular GNN variants (see Table 1).
- **C3 - Sanity checks:** Our sanity checks immediately highlight the strengths and weaknesses of each GNN method before it is sent to production (see Table 2).

**Reproducibility**: Our source code and 'sanity checks' are available at https://bit.ly/3fhWJfK.

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

Table 5: Table of symbols.

| Symbol | Description |
|---|---|
| $G = (\mathbf{A}, \mathbf{X})$ | Undirected graph of $n$ nodes and $d$ features |
| $\mathbf{A}$ | Symmetric adjacency matrix of size $n \times n$ |
| $\mathbf{X}$ | Node feature matrix of size $n \times d$ |
| $\mathbf{y}$ | Node label vector of size $m$, where $m \ll n$ |
| $\tilde{\mathbf{A}} = \mathbf{A} + \mathbf{I}_n$ | Adjacency matrix with self-loops |
| $\tilde{\mathbf{D}} = \mathrm{diag}(\tilde{\mathbf{A}}\mathbf{1}_{n \times 1})$ | Degree matrix of $\tilde{\mathbf{A}}$ |
| $\tilde{\mathbf{A}}_{\mathrm{sym}} = \tilde{\mathbf{D}}^{-1/2}\tilde{\mathbf{A}}\tilde{\mathbf{D}}^{-1/2}$ | Symmetrically normalized $\tilde{\mathbf{A}}$ |
| $\mathbf{A}_{\mathrm{sym}} = \mathbf{D}^{-1/2}\mathbf{A}\mathbf{D}^{-1/2}$ | Symmetrically normalized $\mathbf{A}$ (i.e., no self-loops) |
| $\mathbf{A}_{\mathrm{row}} = \mathbf{D}^{-1}\mathbf{A}$ | Row-wisely normalized $\mathbf{A}$ (i.e., no self-loops) |
| $\mathbf{A}_{\mathrm{col}} = \mathbf{A}\mathbf{D}^{-1}$ | Column-wisely normalized $\mathbf{A}$ (i.e., no self-loops) |
| $\mathrm{diag}(\cdot)$ | Function that creates a diagonal matrix from a vector |
| $\mathbf{1}_{a \times b}$ | Matrix of size $a \times b$ filled with ones |
| $\mathbf{I}_a$ | Identity matrix of size $a \times a$, where the subscript $a$ can be omitted |
| $\mathrm{LR}(\cdot)$ | Logistic regression function defined in Definition 1 |
| $\mathbf{W}$ | Learnable weight matrix of size $h \times c$ in the LR function |
| $\mathcal{P}(\cdot)$ | Feature propagator function defined in Definition 2 |

## A  TABLE OF SYMBOLS

Table 5 summarizes the symbols frequently used in this paper. The formal problem definition and the detailed description of such symbols are presented in Section 2.

## B  PROOF OF LEMMA 1: REPRESENTING LINEAR MODELS WITH GNNLIN

We prove Lemma 1 by representing each linear GNN with GNNLIN. Let $K \geq 0$ be a hyperparameter that determines the number of propagation steps in every GNN.

SGC (Wu et al., 2019) fits the definition of linearization with the following propagator function:

$$\mathcal{P}(\mathbf{A}, \mathbf{X}) = \tilde{\mathbf{A}}_{\mathrm{sym}}^K \mathbf{X}. \tag{4}$$

DGC (Wang et al., 2021) has variants, DGC-Euler and DGC-DK, which have different propagator functions. We focus on DGC-Euler, which is used as the main model in their experiments. DGC is similar to SGC, except that it controls the strength of self-loops as follows:

$$\mathcal{P}(\mathbf{A}, \mathbf{X}) = [(1 - T/K)\mathbf{I} + (T/K)\tilde{\mathbf{A}}_{\mathrm{sym}}]^K \mathbf{X}, \tag{5}$$

where $T > 0$ is a hyperparameter. The self-loops become stronger if $T$ is closer to 0.

S$^2$GC (Zhu & Koniusz, 2021) computes the summation of features propagated with different numbers of steps:

$$\mathcal{P}(\mathbf{A}, \mathbf{X}) = \sum_{k=1}^{K} (\alpha\mathbf{I} + (1 - \alpha)\tilde{\mathbf{A}}_{\mathrm{sym}}^k)\mathbf{X}. \tag{6}$$

The original formulation divides the added features by $K$, which can be safely ignored considering that the weight matrix $\mathbf{W}$ is multiplied to the transformed feature for classification.

G$^2$CN (Li et al., 2022) does not provide an explicit formulation of the propagator function, and thus we derive it. First, the parameterized version $\mathcal{P}'$ of the propagator function is given as follows:

$$\mathcal{P}'(\mathbf{A}, \mathbf{X}; \{\theta_i\}_{i=1}^N) = \sum_{i=1}^{N} \theta_i \mathbf{H}_{i,K}, \tag{7}$$

where $\theta_i$ is a learnable parameter. The $k$-th feature representation $\mathbf{H}_{i,k}$ is recursively defined as

$$
\begin{aligned}
\mathbf{H}_{i,k} &= \mathbf{H}_{i,k-1} - \frac{T_i}{K}(\mathbf{L} - b_i\mathbf{I})^2\mathbf{H}_{i,k-1} \\
&= \mathbf{H}_{i,k-1} - \frac{T_i}{K}((b_i - 1)\mathbf{I} + \mathbf{A}_{\mathrm{sym}})^2\mathbf{H}_{i,k-1} \\
&= [\mathbf{I} - \frac{T_i}{K}((b_i - 1)\mathbf{I} + \mathbf{A}_{\mathrm{sym}})^2]\mathbf{H}_{i,k-1} \\
\mathbf{H}_{i,0} &= \mathbf{X},
\end{aligned}
\tag{8}
$$

where $N$, $T_i$, and $b_i$ are hyperparameters, and $\mathbf{L} = \mathbf{I} - \mathbf{A}_{\mathrm{sym}}$ is the normalized Laplacian matrix. Since the transformed features are combined with a learnable parameter $\theta_i$ in Equation 7, we make a propagator function $\mathcal{P}$ that contains no learnable parameters as follows:

$$
\mathcal{P}(\mathbf{A}, \mathbf{X}) = \overset{N}{\underset{i=1}{\big\|}}\, \mathbf{H}_{i,K} = \overset{N}{\underset{i=1}{\big\|}}\, [\mathbf{I} - \frac{T_i}{K}((b_i - 1)\mathbf{I} + \mathbf{A}_{\mathrm{sym}})^2]^K\mathbf{X}.
\tag{9}
$$

## C  Linearization Processes

We present detailed processes to linearize various graph neural networks (GNN) as in Table 1. The linearization is done in two steps. First, we replace all activation functions with the identity function from the original definition of each GNN. Second, if the resulting layer function contains learnable parameters, we devise a replacement that is at least as expressive as the given function but containing no learnable parameters. This is because our goal of linearization is not just deriving a linear function with respect to $\mathbf{X}$, but understanding GNNs in relation to logistic regression based on our GNNLIN framework. We ignore the bias terms of linear layers for simplicity, without loss of generality.

**Decoupled models**    PPNP, APPNP (Klicpera et al., 2019a), GDC (Klicpera et al., 2019b), and GPR-GNN (Chien et al., 2021) are decoupled GNNs that separate feature transformation and propagation stages. PPNP runs Personalized PageRank on the node features, and APPNP approximates PPNP with $K$ steps of message propagation. GDC generalizes APPNP by increasing the value of $K$ to $\infty$ and sparsifies the propagator matrix $\mathbf{S}$. GPR-GNN also generalizes APPNP to avoid the usage of $\alpha$ by learning a weight for each component, resulting in the concatenation of multiple different features. We provide details of linearization in Appendix C.1.

**Coupled models**    We linearize coupled GNNs including ChebNet (Defferrard et al., 2016), GCN (Kipf & Welling, 2017), GraphSAGE (Hamilton et al., 2017), and GCNII (Chen et al., 2020). The linearized version of GCN is the same as SGC, since the motivation of SGC is to linearize GCN for better scalability and robustness. Although their motivations are different, the linearized versions of ChebNet, GraphSAGE, and GCNII are similar to linearized GPR-GNN in that features propagated by different steps are combined by concatenation. We provide the details in Appendix C.2.

**Attention models**    Attention-based GNNs (Velickovic et al., 2018; Kim & Oh, 2021; Brody et al., 2022) are a popular category of GNNs to learn the importance of each edge based on $\mathbf{X}$ and learnable parameters. Thus, it is not straightforward to linearize them by the GNNLIN framework: even if we assume the learnable parameters in $\mathcal{P}$ as fixed hyperparameters, $\mathcal{P}$ is at least quadratic with $\mathbf{X}$, since $\mathbf{X}$ participates in computing the new adjacency matrix that is multiplied again with $\mathbf{X}$. Nevertheless, we perform incomplete linearization of attention-based models for completeness, and represent the results with the '**' symbol in Table 1. Detailed processes are given in Appendix C.3.

The linearized version of GAT uses a different adjacency matrix for each step $k$, which is the main difference from other linearized models. It is noteworthy that DA-GNN (Liu et al., 2020), proposed as a decoupled model in their paper, is an attention-based model in our analysis. This is because $\mathbf{X}$ participates in computing the new adjacency matrix, making $\mathcal{P}$ quadratic as in GAT.

### C.1  Linearization of Decoupled GNNs

The propagation in decoupled GNNs is done on the abstract representations of node features, which are typically generated by multilayer perceptrons (MLP). Linearization starts with replacing MLPs with linear projections, and removes additional nonlinearity in the process of propagation.

### C.1.1 PPNP (KLICPERA ET AL., 2019A)

The lineraization of PPNP is straightforward, since the authors present a closed-form representation of the propagator function. If we remove the activation function, we have the following:

$$\mathcal{P}(\mathbf{A}, \mathbf{X}) = (\mathbf{I}_n - (1 - \alpha)\tilde{\mathbf{A}}_{\text{sym}})^{-1}\mathbf{X}, \tag{10}$$

where $0 < \alpha < 1$ is a hyperparameter that controls the weight of self-loops.

### C.1.2 APPNP (KLICPERA ET AL., 2019A)

We assume that the initial node representation is created by a single linear layer of $\mathbf{XW}$, where $\mathbf{W}$ is a weight matrix. Then, the $k$-th representation matrix $\mathbf{H}_k$ is represented as follows:

$$\mathbf{H}_k = (1 - \alpha)\tilde{\mathbf{A}}_{\text{sym}}\mathbf{H}_{k-1} + \alpha\mathbf{XW}, \tag{11}$$

where $0 < \alpha < 1$ is a hyperparameter. The closed-form representation of $\mathbf{H}_K$ is given as follows:

$$\mathbf{H}_K = \left([(1 - \alpha)\tilde{\mathbf{A}}_{\text{sym}}]^K + \alpha \sum_{k=0}^{K-1} [(1 - \alpha)\tilde{\mathbf{A}}_{\text{sym}}]^k\right)\mathbf{XW}. \tag{12}$$

We safely remove the weight matrix $\mathbf{W}$, which is redundant, and get the fiinal representation:

$$\mathcal{P}(\mathbf{A}, \mathbf{X}) = [\sum_{k=0}^{K-1} \alpha(1 - \alpha)^k \tilde{\mathbf{A}}_{\text{sym}}^k + (1 - \alpha)^K \tilde{\mathbf{A}}_{\text{sym}}^K]\mathbf{X} \tag{13}$$

### C.1.3 GDC (KLICPERA ET AL., 2019B)

GDC generalizes APPNP and presents various forms of the propagation function. We pick the most representative one given in the paper, which is directly related to APPNP. The unnormalized version of the propagation matrix $\mathbf{S}'$ is given as follows:

$$\mathbf{S}' = \sum_{k=0}^{\infty} \alpha(1 - \alpha)^k \tilde{\mathbf{A}}_{\text{sym}}^k, \tag{14}$$

and then it is normalized and sparsified as

$$\mathbf{S} = \text{sparsify}(\tilde{\mathbf{S}}'_{\text{sym}}). \tag{15}$$

$\tilde{\mathbf{S}}'_{\text{sym}}$ represents adding self-loops and applying the symmetric normalization to $\mathbf{S}'$. The paper gives two approaches for sparsification, which are a) removing elements smaller than $\epsilon$, which is given as a hyperparameter, and b) selecting the top $k$ neighbors for each node. With any choice, the function $\mathcal{P}$ is simply given as follows:

$$\mathcal{P}(\mathbf{A}, \mathbf{X}) = \mathbf{SX}. \tag{16}$$

### C.1.4 GPR-GNN (CHIEN ET AL., 2021)

We assume that the initial node representation is created by a single linear layer of $\mathbf{XW}$, where $\mathbf{W}$ is a weight matrix. Then, we have the following propagator function $\mathcal{P}'$ with parameters:

$$\mathcal{P}'(\mathbf{A}, \mathbf{X}; \{\theta_k\}_{k=0}^K) = \sum_{k=0}^{K} \theta_k \tilde{\mathbf{A}}_{\text{sym}}^k \mathbf{X}, \tag{17}$$

where $\theta_k$ is a parameter that is learned together with $\mathbf{W}$. We replace the summation with concatenation to remove the learnable parameters from $\mathcal{P}'$ and get the following:

$$\mathcal{P}(\mathbf{A}, \mathbf{X}) = \parallel_{k=0}^{K} \tilde{\mathbf{A}}_{\text{sym}}^k \mathbf{X}. \tag{18}$$

## C.2 LINEARIZATION OF COUPLED GNNS

### C.2.1 CHEBNET (DEFFERRARD ET AL., 2016)

Let $\mathbf{H}_k$ be the $k$-th node representation matrix, and $\mathbf{L} = \mathbf{I} - \mathbf{A}_{\text{sym}}$ be the graph Laplacian matrix normalized symmetrically. Then, the propagator function of ChebNet with parameters $\theta$ is

$$\mathcal{P}'(\mathbf{A}, \mathbf{X}; \theta) = \sum_{k=0}^{K-1} \theta_k \mathbf{H}_k, \tag{19}$$

where the recurrence relation is given as follows with the intial terms:

$$\begin{aligned}
\mathbf{H}_0 &= \mathbf{X} \\
\mathbf{H}_1 &= \mathbf{L}\mathbf{X} = -\mathbf{A}_{\text{sym}}\mathbf{X} + \mathbf{X} \\
&\cdots \\
\mathbf{H}_k &= 2(\mathbf{L} - \mathbf{I})\mathbf{H}_{k-1} - \mathbf{H}_{k-2} \\
&= -2\mathbf{A}_{\text{sym}}\mathbf{H}_{k-1} - \mathbf{H}_{k-2}.
\end{aligned} \tag{20}$$

Based on the recurrence relation, the closed-form representation of $\mathbf{H}_k$ is given as

$$\mathbf{H}_k = a_k \mathbf{A}_{\text{sym}}^k \mathbf{X} + a_{k-1} \mathbf{A}_{\text{sym}}^{k-1} \mathbf{X} + \cdots + a_0 \mathbf{X}, \tag{21}$$

where $a_0, \cdots, a_k$ are constants. Since we have $K$ free parameters $\theta_0, \cdots, \theta_K$ corresponding to the $K$ terms in the representation matrix $\mathbf{H}_K$, we safely rewrite the propagator function as

$$\mathcal{P}'(\mathbf{A}, \mathbf{X}; \theta) = \sum_{k=0}^{K-1} \theta_k \mathbf{A}_{\text{sym}}^k \mathbf{X}. \tag{22}$$

Each value of $k$ has a free parameter $\theta_k$. Thus, we generalize it as follows:

$$\mathcal{P}(\mathbf{A}, \mathbf{X}) = \mathop{\Big\|}_{k=0}^{K-1} \mathbf{A}_{\text{sym}}^k \mathbf{X}. \tag{23}$$

### C.2.2 GRAPHSAGE (HAMILTON ET AL., 2017)

We assume the mean aggregator of GraphSAGE. By replacing the activation function as the identity function, each layer $\mathcal{F}$ of GraphSAGE is linearized as follows:

$$\mathcal{F}(\mathbf{X}) = \mathbf{X}\mathbf{W}_1 + \mathbf{A}_{\text{row}}\mathbf{X}\mathbf{W}_2, \tag{24}$$

where $\mathbf{A}_{\text{row}} = \mathbf{D}^{-1}\mathbf{A}$ represents the mean operator in the aggregation, and $\mathbf{W}_1$ and $\mathbf{W}_2$ are learnable weight matrices in the layer. If we apply a chain of two layers, where the weight matrices of the second layer are represented as $\mathbf{W}_3$ and $\mathbf{W}_4$, we get the following:

$$\mathcal{F}(\mathcal{F}(\mathbf{X})) = \mathbf{X}\mathbf{W}_1\mathbf{W}_3 + \mathbf{A}_{\text{row}}\mathbf{X}(\mathbf{W}_1\mathbf{W}_2 + \mathbf{W}_2\mathbf{W}_3) + \mathbf{A}_{\text{row}}^2\mathbf{X}\mathbf{W}_2\mathbf{W}_4 \tag{25}$$

$$= \mathbf{X}\mathbf{W}_a + \mathbf{A}_{\text{row}}\mathbf{X}\mathbf{W}_b + \mathbf{A}_{\text{row}}^2\mathbf{X}\mathbf{W}_c, \tag{26}$$

where we redefine the weight matrices without loss of generality as

$$\mathbf{W}_a = \mathbf{W}_1\mathbf{W}_3 \tag{27}$$

$$\mathbf{W}_b = \mathbf{W}_1\mathbf{W}_2 + \mathbf{W}_2\mathbf{W}_3 \tag{28}$$

$$\mathbf{W}_c = \mathbf{W}_2\mathbf{W}_4. \tag{29}$$

If we generalize it into $K$ layers, we get the following:

$$\mathcal{F}^K(\mathbf{X}) = \sum_{k=1}^{K} \mathbf{A}_{\text{row}}^k \mathbf{X}\mathbf{W}_k. \tag{30}$$

Note that a different weight matrix $\mathbf{W}_k$ is applied to each layer $k$. This is equivalent to concatenating the transformed features of all layers and learning a single large weight matrix in training.

### C.2.3 GCNII (CHEN ET AL., 2020)

After replacing the activation function with the identity function, the $l$-th layer $\mathcal{F}_l$ of GCNII is given as follows:

$$\mathcal{F}_l(\mathbf{H}) = ((1 - \alpha_l)\tilde{\mathbf{A}}_{\text{sym}}\mathbf{H} + \alpha_l\mathbf{X}))((1 - \beta_l)\mathbf{I} + \beta_l\mathbf{W}_l), \tag{31}$$

where $\alpha_l$ and $\beta_l$ are hyperparameters, and $\mathbf{W}_l$ is a weight matrix. The second term is equivalent to $\mathbf{W}_l$ regardless of the value of $\beta_l$, since $\mathbf{W}_l$ is a free parameter. We also set $\alpha_l$ to a constant $\alpha$ which is the same for every layer $l$, following the original paper (Chen et al., 2020).[1] Then, the equation is simplified as

$$\mathcal{F}_l(\mathbf{H}) = ((1 - \alpha)\tilde{\mathbf{A}}_{\text{sym}}\mathbf{H} + \alpha\mathbf{X})\mathbf{W}_l. \tag{32}$$

If we apply a chain of two layers $l$ and $l + 1$, we get the following:

$$\mathcal{F}_{l+1}(\mathcal{F}_l(\mathbf{H})) = ((1 - \alpha)\tilde{\mathbf{A}}_{\text{sym}}((1 - \alpha)\tilde{\mathbf{A}}_{\text{sym}}\mathbf{H} + \alpha\mathbf{X})\mathbf{W}_l + \alpha\mathbf{X})\mathbf{W}_{l+1} \tag{33}$$

$$= (1 - \alpha)^2\tilde{\mathbf{A}}_{\text{sym}}^2\mathbf{H}\mathbf{W}_l\mathbf{W}_{l+1} + \alpha(1 - \alpha)\tilde{\mathbf{A}}_{\text{sym}}\mathbf{X}\mathbf{W}_l\mathbf{W}_{l+1} + \alpha\mathbf{X}\mathbf{W}_{l+1} \tag{34}$$

$$= (1 - \alpha)^2\tilde{\mathbf{A}}_{\text{sym}}^2\mathbf{H}\mathbf{W}_l' + \alpha(1 - \alpha)\tilde{\mathbf{A}}_{\text{sym}}\mathbf{X}\mathbf{W}_l' + \alpha\mathbf{X}\mathbf{W}_{l+1}, \tag{35}$$

where $\mathbf{W}_l' = \mathbf{W}_l\mathbf{W}_{l+1}$, which is also a free parameter. If we generalize it into $K$ layers, we get the following:

$$\mathcal{F}^K(\mathbf{X}) = (1 - \alpha)^{K-1}((1 - \alpha)\tilde{\mathbf{A}}_{\text{sym}}^K + \alpha\tilde{\mathbf{A}}_{\text{sym}}^{K-1})\mathbf{X}\mathbf{W}_K + \alpha\sum_{k=0}^{K-2}(1 - \alpha)^k\tilde{\mathbf{A}}_{\text{sym}}^k\mathbf{X}\mathbf{W}_k. \tag{36}$$

We safely remove the constant from each term, which can be included in the weight matrix:

$$\mathcal{F}^K(\mathbf{X}) = ((1 - \alpha)\tilde{\mathbf{A}}_{\text{sym}}^K + \alpha\tilde{\mathbf{A}}_{\text{sym}}^{K-1})\mathbf{X}\mathbf{W}_K + \sum_{k=0}^{K-2}\tilde{\mathbf{A}}_{\text{sym}}^k\mathbf{X}\mathbf{W}_k. \tag{37}$$

We replace the summation operators between terms having different weight matrices with concatenation operators, having the final propagator function $\mathcal{P}$ as follows:

$$\mathcal{P}(\mathbf{A}, \mathbf{X}) = \overset{K-2}{\underset{k=0}{\Big\|}} \tilde{\mathbf{A}}_{\text{sym}}^k\mathbf{X} \,\|\, ((1 - \alpha)\tilde{\mathbf{A}}_{\text{sym}}^K + \alpha\tilde{\mathbf{A}}_{\text{sym}}^{K-1})\mathbf{X} \tag{38}$$

### C.3 LINEARIZATION OF ATTENTION GNNS

### C.3.1 DA-GNN (LIU ET AL., 2020)

We assume that the initial node representation is created by a single linear layer of $\mathbf{X}\mathbf{W}$, where $\mathbf{W}$ is a weight matrix. Then, the $k$-th representation matrix $\mathbf{H}_k$ is represented as follows:

$$\mathbf{H}_k = \tilde{\mathbf{A}}_{\text{sym}}^k\mathbf{X}\mathbf{W}. \tag{39}$$

DA-GNN computes the weighted sum of representations for all $k \in [0, K]$, where the weight values are determined also from the representation matrices:

$$\begin{aligned}\mathcal{P}(\mathbf{A}, \mathbf{X}) &= \sum_{k=0}^{K}\text{diag}(\mathbf{H}_k\mathbf{s})\mathbf{H}_k \\ &= \sum_{k=0}^{K}\text{diag}(\tilde{\mathbf{A}}_{\text{sym}}^k\mathbf{X}\mathbf{W}\mathbf{s})\tilde{\mathbf{A}}_{\text{sym}}^k\mathbf{X}\mathbf{W}.\end{aligned} \tag{40}$$

where $\mathbf{s}$ is a learnable weight vector. We safely remove the last $\mathbf{W}$ and rewrite $\mathbf{W}\mathbf{s}$ as $\mathbf{w}$. Then, we have the final representation of the propagator function:

$$\mathcal{P}(\mathbf{A}, \mathbf{X}) = \sum_{k=0}^{K}\text{diag}(\tilde{\mathbf{A}}_{\text{sym}}^k\mathbf{X}\mathbf{w})\tilde{\mathbf{A}}_{\text{sym}}^k\mathbf{X}. \tag{41}$$

---

[1]$\alpha$ is set to 0.1 in the original paper of GCNII (Chen et al., 2020).

### C.3.2    GAT (VELICKOVIC ET AL., 2018)

We apply the following changes to linearize GAT, whose linearization is not straightforward due to the nonlinearity in the attention function:

1. We replace the activation functions between layers with the identity functions.
2. We simplify the attention function $\alpha_{ij} = \exp(e_{ij})/\sum_k \exp(e_{ik})$ as $e_{ij}$.
3. We remove the LeaklyReLU function in the computation of $e_{ij}$.
4. We assume the single-head attention.

The edge weight $e_{ij}$, which is the $(i, j)$-th element of the propagator matrix, is defined as follows:

$$e_{ij} = \mathbf{a}_{\text{dst}}^\top(\mathbf{W}^\top \mathbf{x}_i) + \mathbf{a}_{\text{src}}^\top(\mathbf{W}^\top \mathbf{x}_j), \tag{42}$$

where $\mathbf{x}_i$ and $\mathbf{x}_j$ are feature vectors of length $d$ for node $i$ and $j$, respectively, $\mathbf{W}$ is a $d \times c$ learnable weight matrix, and $\mathbf{a}_{\text{dst}}$ and $\mathbf{a}_{\text{src}}$ are learnable weight vectors of length $c$. Then, we derive the initial form of a linearized GAT layer as follows:

$$\mathbf{H} = [\text{diag}(\mathbf{XWa}_{\text{dst}})\tilde{\mathbf{A}} + \tilde{\mathbf{A}}\text{diag}(\mathbf{XWa}_{\text{src}})]\mathbf{XW}. \tag{43}$$

Since all $\mathbf{a}_{\text{dst}}$, $\mathbf{a}_{\text{src}}$, and $\mathbf{W}$ are free parameters, we generalize it as follows:

$$\mathbf{H} = [\text{diag}(\mathbf{Xw}_{\text{dst}})\tilde{\mathbf{A}} + \tilde{\mathbf{A}}\text{diag}(\mathbf{Xw}_{\text{src}})]\mathbf{XW}, \tag{44}$$

where $\mathbf{w}_{\text{dst}}$ and $\mathbf{w}_{\text{src}}$ are learnable vectors of length $m$ that replace $\mathbf{a}_{\text{dst}}$ and $\mathbf{a}_{\text{src}}$, respectively.

## D    DETAILS ON SANITY CHECKS

### D.1    FORMAL DEFINITIONS OF GRAPH SCENARIOS

We categorize all possible scenarios of node classification based on the characteristics of node features $\mathbf{X}$, a graph structure $\mathbf{A}$, and node labels $\mathbf{y}$. We denote by $A_{ij}$ and $Y_i$ the random variables for edge $(i, j)$ between nodes $i$ and $j$ and label $y_i$ of node $i$, respectively.

**Edges**    For the adjacency matrix $\mathbf{A}$, we consider the following three cases:

- **Uniform:** $P(A_{ki} = 1 \mid A_{ij} = A_{jk} = 1) = P(A_{ki} = 1)$
- **Clustered:** $P(A_{ki} = 1 \mid A_{ij} = A_{jk} = 1) > P(A_{ki} = 1)$
- **Bipartite:** $P(A_{ki} = 1 \mid A_{ij} = A_{jk} = 1) < P(A_{ki} = 1)$

The uniform case represents that every edge is determined independently of the others, and the graph structure gives no useful information for node classification. In the clustered case, it is more likely that nodes having common neighbors make more connections in a graph. In the bipartite case, nodes make more connections with those sharing no common neighbors.

**Labels**    For the labels in $\mathbf{y}$, we consider the following three cases:

- **Individual:** $P(Y_i = y \mid A_{ij} = 1, Y_j = y) = P(Y_i = y)$
- **Homophily:** $P(Y_i = y \mid A_{ij} = 1, Y_j = y) > P(Y_i = y)$
- **Heterophily:** $P(Y_i = y \mid A_{ij} = 1, Y_j = y) < P(Y_i = y)$

The individual case means that the label of a node is independent of the labels of its neighbors. This is the case when the graph structure works as noise information with regard to classification. In the homophily case, adjacent nodes are likely to have the same label. It is the most popular assumption of GNNs for node classification. In the heterophily case, adjacent nodes are likely to have different labels, which is not as common as homophily but often observed in real-world graphs.

**Features**    For the feature matrix $\mathbf{X}$, we consider the following three cases based on $\mathbf{A}$ and $\mathbf{y}$. We use the notation of $p(\cdot)$ since the features are typically modeled as continuous variables.

- **Random:** $p(\mathbf{x}_i, \mathbf{x}_j \mid y_i, y_j, a_{ij}) = p(\mathbf{x}_i, \mathbf{x}_j)$
- **Structural:** $p(\mathbf{x}_i, \mathbf{x}_j \mid y_i, y_j, a_{ij}) \neq p(\mathbf{x}_i, \mathbf{x}_j \mid y_i, y_j)$

- **Semantic:** $p(\mathbf{x}_i, \mathbf{x}_j \mid y_i, y_j, a_{ij}) \neq p(\mathbf{x}_i, \mathbf{x}_j \mid a_{ij})$

The random case represents that features are determined independently of the graph and labels. In this case, node features do not give useful information for classification, but work as unique indices of nodes like one-hot embeddings if the dimensionality $d$ of features is large. In the structural case, features are correlated with the graph structure, but not directly with labels. Such features give useful information for classification only if the graph structure and labels are related. In the semantic case, features directly provide useful information for predicting labels. Note that features can be semantic and structural at the same time by satisfying both of the conditions.

### D.2    IMPLEMENTATION OF GRAPH SCENARIOS

There are various ways to generate synthetic graphs satisfying the definitions of different scenarios. One can use a synthetic graph generator designed to create more plausible graphs (Leskovec et al., 2010; Barabási & Albert, 1999), but we choose the simplest one to focus on the mutual information, rather than the other characteristics of synthetic graphs such as the degree distribution.

**Structure**    We assume that the number of structural clusters is the same as the number $c$ of labels for the alignment with label information. We divide all nodes into $c$ groups and then decide the edge densities for intra- and inter-connections of groups based on the structural type: uniform, clustered, and bipartite. We use a hyperparameter $\epsilon_a$ to determine the noise level in the case of homophily or heterophily: for example, if $\epsilon_a = 0$, the graph has a full block-diagonal adjacency matrix in the case of homophily. The expected number of edges is the same for all three cases, and we set $\epsilon_a = 0$ in all of our experiments.

A notable characteristic of a bipartite structure $\mathbf{A}$ is that $\mathbf{A}^2$ has a clustered structure. If we create inter-group connections for every possible pair of different groups, even noise-free $\mathbf{A}$ with $\epsilon_a = 0$ makes noisy $\mathbf{A}^2$ with inter-group connections. For better consistency, we set the number of classes to an even number in our experiments, randomly pick paired classes such as $(1, 3)$ and $(2, 4)$ when $c = 4$, for example, and create inter-group connections only for the chosen pairs. In this way, we create a non-diagonal block-permutation matrix $\mathbf{A}$ when $\epsilon_a = 0$, and the noise level of $\mathbf{A}^2$, which has a clustered structure, is solely controlled by the noise level of $\mathbf{A}$.

**Labels**    In the case of individual $\mathbf{y}$, we determine the label of each node uniformly at random, with no consideration of the graph structure. In the case of homophily or heterophily $\mathbf{y}$, we assign labels based on the groups of nodes assumed by the graph structure. That is, nodes in the same group have the same label, and thus the group index itself works as the label $\mathbf{y}$. In this way, we force homophily $\mathbf{y}$ for clustered $\mathbf{A}$, and heterophily $\mathbf{y}$ for bipartite $\mathbf{A}$. The degree of homophily (or heterophily) is also determined by the noise level $\epsilon_a$ of the graph structure.

**Features**    We basically assume that every feature element is sampled from a uniform distribution. Thus, in the random case, we sample each element from the uniform distribution $\mathcal{U}(0, 1)$ between 0 and 1. In the structural case, we run low-rank support vector decomposition (SVD) (Halko et al., 2011) to make $\mathbf{X}$ have structural information. Given $\mathbf{U\Sigma V}^\top \approx \mathbf{A}$ from low-rank SVD, we take $\mathbf{U}$ and normalize each feature element to have the zero-mean and unit-variance. The rank $r$ in the SVD is determined as a hyperparameter; higher $r$ captures the structure better, but can give noisy information. We also apply the ReLU function to $\mathbf{U}$ to make them positive.

In the semantic case, we randomly pick $c$ representative vectors $\{\mathbf{v}_k\}_{k=1}^c$ from the uniform distribution, which correspond to the $c$ different classes. Then, for each node $i$ with label $y$, we sample a feature vector such that $\arg\max_k \mathbf{x}^\top \mathbf{v}_k = y$. In this way, we have random vectors having sufficient semantic information for the classification of labels, with a guarantee that the perfect linear decision boundaries can be drawn in the feature space $\mathbf{X}$ at the training time.

## E    REPRODUCIBILITY

### E.1    DATASETS

We use 7 homophily and 6 heterophily datasets in experiments, which were used widely in previous works on node classification (Chien et al., 2021; Pei et al., 2020). Cora, CiteSeer, and PubMed (Sen et al., 2008; Yang et al., 2016) are homophily citation graphs between research articles. Computers

Table 6: **The statistics of datasets** used in our experiments. The first seven datasets are homophily graphs, while the last six are heterophily graphs.

|  | Cora | CiteSeer | PubMed | Computers | Photo | ogbn-arXiv | ogbn-Products | Chameleon | Squirrel | Actor | Penn94 | Twitch | Pokec |
|---|---|---|---|---|---|---|---|---|---|---|---|---|---|
| # of Nodes | 2.7K | 3.3K | 19.7K | 13.8K | 7.7K | 169K | 2.4M | 2.3K | 5.2K | 7.6K | 41.6K | 168K | 1.6M |
| # of Edges | 5.3K | 4.6K | 44.3K | 246K | 119K | 1.2M | 61.9M | 31.4K | 198K | 26.7K | 1.4M | 6.8M | 30.6M |
| # of Features | 1433 | 3703 | 500 | 767 | 745 | 128 | 100 | 2325 | 2089 | 931 | 4814 | 7 | 65 |
| # of Classes | 7 | 6 | 3 | 10 | 8 | 40 | 39 | 5 | 5 | 5 | 2 | 2 | 2 |

Table 7: **Search spaces of hyperparameters.**

| Method | Hyperparameters |
|---|---|
| LR | $wd = [0, 5e^{-4}]$ |
| SGC | $wd = [0, 5e^{-4}], K = 2$ |
| DGC | $wd = [0, 5e^{-4}], K = 200, T = [3, 4, 5, 6]$ |
| S$^2$GC | $wd = [0, 5e^{-4}], K = 16, \alpha = [0.01, 0.03, 0.05, 0.07, 0.09]$ |
| G$^2$CN | $wd = [0, 5e^{-4}], K = 100, N = 2, T_1 = T_2 = [10, 20, 30, 40], b_1 = 0, b_2 = 2$ |
| GCN | $wd = [0, 5e^{-4}], lr = [2e^{-3}, 0.01, 0.05], K = 2$ |
| SAGE | $wd = [0, 5e^{-4}], lr = [2e^{-3}, 0.01, 0.05], K = 2$ |
| GCNII | $wd = [0, 5e^{-4}], lr = 0.01, K = [8, 16, 32, 64], \alpha = [0.1, 0.2, 0.5], \theta = [0.5, 1, 1.5]$ |
| APPNP | $wd = [0, 5e^{-4}], lr = [2e^{-3}, 0.01, 0.05], K = 10, \alpha = 0.1$ |
| GPR-GNN | $wd = [0, 5e^{-4}], lr = [2e^{-3}, 0.01, 0.05], K = 10, \alpha = [0.1, 0.2, 0.5, 0.9]$ |
| GAT | $wd = [0, 5e^{-4}], lr = [2e^{-3}, 0.01, 0.05], K = 2, heads = 8$ |
| *SlenderGNN* | $wd_1 = [1e^{-3}, 1e^{-4}, 1e^{-5}], wd_2 = [1e^{-3}, 1e^{-4}, 1e^{-5}, 1e^{-6}]$ |

and Photo (Shchur et al., 2018) are homophily Amazon co-purchase graphs between items. ogbn-arXiv and ogbn-Products are large homophily graphs from Open Graph Benchmark (Hu et al., 2020). Since we use only 2.5% of all labels as training data, we omit the classes with instances fewer than 100. Chameleon and Squirrel (Rozemberczki et al., 2021) are heterophily Wikipedia graphs. Actor (Tang et al., 2009) is a heterophily graph connected by co-occurrence of actors on Wikipedia pages. Penn94 (Traud et al., 2012; Lim et al., 2021) is a heterophily graph of gender relations in a social network. Twitch (Rozemberczki & Sarkar, 2021) and Pokec (Leskovec & Krevl, 2014) are large graphs, which have been relabeled by (Lim et al., 2021) to be heterophily. We make the heterophily graphs undirected as done in (Chien et al., 2021). The statistics of datasets are reported in Table 6.

### E.2 COMPETITORS

The propagator functions of graph kernel methods (Smola & Kondor, 2003) are given as follows:

$$(\text{Reg. Kernel}) \; \mathcal{P}(\mathbf{A}, \mathbf{X}) = (\mathbf{I}_n + \sigma^2 \tilde{\mathbf{L}})^{-1} \mathbf{X} \tag{45}$$

$$(\text{Diff. Kernel}) \; \mathcal{P}(\mathbf{A}, \mathbf{X}) = \exp(-\sigma^2/2\tilde{\mathbf{L}})\mathbf{X} \tag{46}$$

$$(\text{RW Kernel}) \; \mathcal{P}(\mathbf{A}, \mathbf{X}) = (a\mathbf{I}_n - \tilde{\mathbf{L}})^p \mathbf{X}, \tag{47}$$

where $\tilde{\mathbf{L}} = \mathbf{D}^{-1/2}(\mathbf{D} - \mathbf{A})\mathbf{D}^{-1/2}$ is the normalized Laplacian matrix, and $\sigma = 1$, $a = 1$, and $p = 2$ are hyperparameters. We use the reasonable default values introduced in the paper.

We perform row-normalization on the node features of all datasets as done in most studies on GNNs. We report the hyperparameters used for a grid search in Table 7, which is done for every split of data. The dimensions of hidden layers are all set to 64, and the probabilities of dropout layers are all set to 0.5. For the linear models, we use L-BFGS as the optimizer for training 100 epochs with patience 5; for the nonlinear ones, we use ADAM and train them for 1000 epochs with patience 200.

*SlenderGNN* contains only 2 hyperparameters, where $wd_1$ is the weight of LASSO, and $wd_2$ is the weight of group LASSO. It is worth noting that, when searching the hyperparameters, *SlenderGNN* does not need to recompute the features, while most of the linear methods need to do so because of including one or more hyperparameters in the features that can be precomputed otherwise.

Table 8: **Ablation Study - 2-step aggregation is good enough:** the first three places are statistically tied. Green (■, ■, ■) marks the top three (higher is darker).

| $k_{row}$ | $k_{sym}$ | Cora | CiteSeer | PubMed | Comp. | Photo | ArXiv | Products | Cham. | Squirrel | Actor | Penn94 | Twitch | Pokec |
|---|---|---|---|---|---|---|---|---|---|---|---|---|---|---|
| 2 | 2 | 77.8±1.1 | 67.1±2.3 | 84.6±0.5 | 86.3±0.7 | 91.8±0.5 | 66.3±0.3 | 84.9±0.0 | 40.8±3.2 | 31.1±0.7 | 30.9±0.6 | 68.2±0.6 | 59.7±0.1 | 73.9±0.1 |
| 4 | 3 | 79.2±0.8 | 66.1±3.5 | 84.2±0.5 | 85.8±0.6 | 91.3±0.4 | 67.2±0.1 | 85.4±0.0 | 38.6±7.1 | 29.5±1.8 | 30.4±0.7 | 67.5±0.6 | 60.0±0.1 | 74.6±0.1 |
| 6 | 4 | 79.2±0.7 | 66.2±3.4 | 84.0±0.3 | 85.2±0.7 | 91.0±0.8 | 67.3±0.2 | 84.7±1.7 | 38.2±6.3 | 29.2±1.5 | 31.2±0.8 | 67.7±0.7 | 59.8±0.1 | 74.2±0.2 |
| [2, 4] | [2, 3] | 78.9±0.9 | 66.9±2.5 | 84.0±0.5 | 86.2±0.7 | 91.5±0.5 | 67.4±0.1 | 73.9±23. | 41.4±4.0 | 31.0±0.7 | 30.4±0.4 | 68.3±0.5 | 60.8±0.1 | 76.0±0.1 |
| [2, 4, 6] | [2, 3, 4] | 79.4±1.1 | 66.0±4.4 | 84.4±0.4 | 85.9±0.5 | 91.3±0.5 | 67.9±0.2 | 84.8±2.1 | 41.0±3.9 | 31.4±0.6 | 29.8±0.6 | 68.2±0.6 | 60.7±0.1 | 76.6±0.2 |

Table 9: **Ablation Study - *SlenderGNN* is minimalistic:** all its components are needed to achieve good accuracy in real-world graphs. The first two places are highlighted.

| Model | Cora | CiteSeer | PubMed | Comp. | Photo | ArXiv | Products | Cham. | Squirrel | Actor | Penn94 | Twitch | Pokec |
|---|---|---|---|---|---|---|---|---|---|---|---|---|---|
| w/o Sp. Reg. | **77.8±0.6** | 65.0±3.5 | 83.8±0.5 | 85.9±0.8 | 91.7±0.7 | 65.2±0.2 | 83.4±1.9 | 40.1±3.8 | 30.7±1.0 | 30.1±0.6 | 67.4±0.6 | **59.8±0.1** | 74.2±0.0 |
| w/o PCA | 74.8±1.5 | 66.0±3.1 | **84.7±0.5** | 84.4±1.1 | 90.3±0.7 | 60.8±0.2 | **84.5±0.0** | **41.3±2.0** | **31.8±1.1** | 27.3±1.1 | **67.7±0.7** | 59.1±0.2 | 72.8±0.1 |
| w/o **U** | **78.1±1.0** | **67.4±2.5** | 84.4±0.3 | **86.0±0.5** | **92.1±0.4** | **66.1±0.2** | 82.5±0.6 | 37.5±4.2 | 29.8±0.4 | **31.3±0.5** | 65.8±0.6 | 56.8±0.0 | 72.8±0.1 |
| *SlenderGNN* | 77.8±1.1 | 67.1±2.3 | 84.6±0.5 | 86.3±0.7 | 91.8±0.5 | 66.3±0.3 | 84.9±0.0 | 40.8±3.2 | 31.1±0.7 | 30.9±0.6 | 68.2±0.6 | 59.7±0.1 | 73.9±0.1 |

# F    ABLATION STUDIES

**Nonlinearity**    Adding nonlinearity does not necessarily improve the accuracy, since the increased    ← **R-f1, y1, y2, D2**
difficulty of training and hyperparameter tuning makes the model less generalizable. To demonstrate
this, we design three nonlinear variants of *SlenderGNN*:

- **w/ MLP-2:** We replace LR with a 2-layer MLP.
- **w/ MLP-3:** We replace LR with a 3-layer MLP.
- **w/ NL Trans.:** We replace the PCA function $g(\cdot)$ with a nonlinear function. Specifically, we adopt a 2-layer MLP for the first two components, and a 2-layer GCN for the last two. The transformed features are concatenated and input into another 2-layer MLP.

We use dropout with a probability $0.5$ to prevent overfitting in both MLP and GCN. The nonlinear models are trained with the same setting as GCN reported in Table 7. We report the result in Table 4, showing that adding nonlinearity does not necessarily improve the accuracy, while sacrificing both scalability and interpretability. We choose to use linear function based on this result.

**Receptive Fields**    To test the effect of changing the receptive field of our *SlenderGNN*, we vary the    ← **R-V1**
distance of aggregation as in Table 8. $k_{row}$ denotes the number of steps for $\mathbf{A}_{row}$, while $k_{sym}$ denotes
the number of steps for $\tilde{\mathbf{A}}_{sym}$. Since $\mathbf{A}_{row}$ is designed to consider heterophily relations, we use only
the even values of $k_{row}$. Table 8 shows that the first three places in each dataset are statistically tied
in most of the cases. In other words, we have no significant gain by increasing the values of $k_{row}$ and
$k_{sym}$, and thus the 2-step aggregation for both $\mathbf{A}_{row}$ and $\tilde{\mathbf{A}}_{sym}$ is good enough. To keep our model
simple and general, we use $k_{row} = 2$ and $k_{sym} = 2$ in all experiments on synthetic and real data.

**Components**    We evaluate the accuracy of *SlenderGNN* when each of its core modules is disabled:
the sparse regularization, PCA, and structural features $\mathbf{U}$. *SlenderGNN* performs well with all these
ideas in all 13 datasets, i.e., it is always included in the top two at each dataset. This shows that our
*SlenderGNN* is designed effectively with ideas that help improve its performance.

