# OpenReview forum: "SlenderGNN: Accurate, Robust, and Interpretable GNN, and the Reasons for its Success"
_ICLR.cc/2023/Conference — Submitted to ICLR 2023_

### Official Review · Reviewer_DLLT · 2022-10-24

**Confidence:** 3
**Correctness:** 1
**Technical Novelty And Significance:** 1
**Empirical Novelty And Significance:** 1
**Recommendation:** 3

**Clarity, Quality, Novelty And Reproducibility:**

- Clarity:
In general, the way the proposed method works (section 4 of the paper) is not very clear. e.g. The connection between the components of the propagation function and GNNLin (Table 1) is not clear and the choices of 2-step neighbors, neighbors, etc. is not well justified in equation 3 (i.e. propagation function).

- Quality
Interpretability and robustness are not well explained in the paper and they are not backed by experimental results either. e.g. there are quantification methods for interpretability/explainability in the literature that authors could have reported for their work. For example, one metric is to have a data set with explanation ground truth and measure the overlap between the decisioning factor identified by the proposed method vs. the ground truth explanation. Same point holds about robustness where authors claims are not backed by strong empirical evidence.

- Novelty
Novelty contributions are marginal.

- Reproducibility
Although authors have provided the source code for their method, due to lack of clarity explained earlier in this box, it's hard to reproduce authors work.

**Strength And Weaknesses:**

Strength
- The paper addresses an important topic (i.e. interpretability and robustness of GNN).

Weaknesses
- Scattered and not well supported claims. The contributions from GNNLin and SlenderGNN are mixed in the paper and empirical results only provide partial evidence about the claims. Maybe it would have been more beneficial if authors had fully focused on SlenderGNN rather than including SlenderGNN related materials as well.
- Experimental results are not clearly discussed and it is not clear why LR shows comparable to superior results in most data sets vs. more state of the art and GNN-based approaches.
- Also refer to reviewer's comment on clarity, quality, and novelty in the box below.

**Summary Of The Paper:**

The paper proposes SlenderGNN, a new GNN method that is accurate, robust, and interpretable. Authors report experimental results comparing SlenderGNN's performance with a set of alternative methods. They also conduct an ablated study to assess the importance of individual modules of the proposed method in its performance.

**Summary Of The Review:**

The paper addresses an important question/concern with GNNs, however there are major limitations in connecting the experimental results with the paper's claims. Also theoretical grounds of the paper are not very clear.

---

> ### Author Response · Authors · 2022-11-17
> **Response to Reviewer DLLT**
>
> We thank the reviewer for the detailed feedback. We respond to the concerns raised by the reviewer.
>
> > **R-D1 (Claims and contributions):** Scattered and not well supported claims. The contributions from GNNLin and SlenderGNN are mixed in the paper and empirical results only provide partial evidence about the claims. Maybe it would have been more beneficial if authors had fully focused on SlenderGNN rather than including SlenderGNN related materials as well.
>
> **Claims:** We updated the paper to clarify our claims and the evidence for such claims. In Section 4.1 and 4.2, we justified our design decisions from the pain points learned from GNNLin. In Section 6, we first presented our research questions and answered them through experiments on real-world graphs. We also performed additional ablation studies to justify our design decisions  empirically (see Table 4 and 8).
>
> **Contributions:** We believe that our two other contributions, GNNLin and the sanity checks, are important to understand how SlenderGNN succeeds even with its simplicity. GNNLin allows us to easily compare many existing models and to find their pain points, and the sanity checks demonstrate such pain points through experiments on controlled synthetic graphs. The insights from GNNLin and the sanity checks are essential motivations of SlenderGNN.
>
> > **R-D2 (Discussion):** Experimental results are not clearly discussed and it is not clear why LR shows comparable to superior results in most data sets vs. more state of the art and GNN-based approaches.
>
> We added in-depth discussion and additional experimental results throughout the updated paper to better explain the success of SlenderGNN. Please see our new results in Section 6, especially our new ablation study in Table 4 that compares SlenderGNN with its nonlinear variants.
>
> > **R-D3 (Clarity):** Clarity: In general, the way the proposed method works (section 4 of the paper) is not very clear. e.g. The connection between the components of the propagation function and GNNLin (Table 1) is not clear and the choices of 2-step neighbors, neighbors, etc. is not well justified in equation 3 (i.e. propagation function).
>
> We improved the justification of the four components of our SlenderGNN based on the pain points of existing models. Please see Section 4.1 and 4.2 in the updated paper, and our new ablation studies in Section 6.
>
> > **R-D4 (Interpretability and robustness):** Quality Interpretability and robustness are not well explained in the paper and they are not backed by experimental results either. e.g. there are quantification methods for interpretability/explainability in the literature that authors could have reported for their work. For example, one metric is to have a data set with explanation ground truth and measure the overlap between the decisioning factor identified by the proposed method vs. the ground truth explanation. Same point holds about robustness where authors claims are not backed by strong empirical evidence.
>
> **Interpretability:** Thank you for the suggestion. We’d like to clarify that we did a similar experiment to show the interpretability of SlenderGNN in the submitted version of our paper. If you check Figure 2 in the updated paper, we compare the learned weights of SlenderGNN to the ground truth knowledge in the sanity checks where we know the answer of explanation.
>
> **Robustness:** In this work, we use the term robustness to refer to the ability of a model to be able to work with different properties of graphs. SlenderGNN succeeds in every sanity check in Table 2, while none of the existing models does, and performs very well in 13 different real-world datasets in Table 3. As we emphasized in the paper, SlenderGNN is the only method without red cells in both Table 2 and 3. In fact, the robustness (and the generalizability) of SlenderGNN is the main reason that it outperforms more sophisticated and complex ones in real-world graphs despite its simplicity.
>
> > **R-D5 (reproducibility):** Reproducibility Although authors have provided the source code for their method, due to lack of clarity explained earlier in this box, it's hard to reproduce authors work.
>
> Our code is designed to reproduce the results in the paper. You can download our code (via https://bit.ly/3fhWJfK), install the required packages (pip install -r requirements.txt), and run the main script (python src/main.py) to reproduce the results. All the datasets are publicly available and will be automatically downloaded before training the model.

---

> ### Author Response · Authors · 2022-11-29
> **Looking forward to your response**
>
> Dear reviewer,
>
> Thank you again for your thoughtful review. Based on your suggestions, we improved the overall presentation of our paper and clarified the strengths of SlenderGNN. We also included new results to support the strengths more clearly. We would love to hear your thoughts on our response. Please let us know if there is anything else we can do to address your comments.

---

### Official Review · Reviewer_y6fQ · 2022-10-29

**Confidence:** 4
**Correctness:** 4
**Technical Novelty And Significance:** 2
**Empirical Novelty And Significance:** 3
**Recommendation:** 5

**Clarity, Quality, Novelty And Reproducibility:**

- The manuscript is well-written for authors to follow easily.
- The model is well-organized, but it is actually a logistic regression with features. In that sense, the proposed model is not novel enough to this literature.
- Sanity checking framework is great to assess the model performance in various scenarios.

**Strength And Weaknesses:**

* Strengths
- A simple but generic interpretable model is proposed.
- The proposed model achieves many aspects all together.
- Sanity check framework is proposed and it would be great for other researchers to analyze their own models.

* Weaknesses
- The non-linear component is cleared for simplicity, but it is not clear whether the activation is not necessary or it can play a role to improve the performance more.
- In particular, the proposed method is just a logistic regression model that uses neighbor information, connectivity information, and node attributes as features. Eventually, the proposed model solves the iid logistic regression regardless of any latent space. Apparently, the proposed model does not seem to be a GNN. Authors need to modify the definition or the explanation of modeling, or regard the proposed model as the linear model using graph features. Robustness and interpretability can be achieved typically through such simple models.
- Many GNN models rely on sampling methods, while the proposed method is based on more matrix operation. It would be great if the proposed method can work with some sampled data or scale enough.


**Summary Of The Paper:**

Authors propose the framework that tackles crucial aspects of modeling, which include accuracy, robustness, slenderness, and interpretability. To tackle this, authors propose the generic framework, named GNNLIN, that consists of the propagation layer and the logistic regression layer. Through the generically structured propagation layer, the proposed method can provide the interpretability. Furthermore, authors propose the sanity checking framework that verifies whether a given model algorithms works broadly for various scenarios of graphs and features. Finally, authors use the proposed sanity checks and real-world datasets to show the proposed algorithm outperforming the baseline methods.

**Summary Of The Review:**

The proposed framework is very intuitive and passes the sanity checking framework as well as outperforms the baseline methods. However, the model is essentially the logistic regression model using graph and node features, which are already being used commonly by industry. The novelty of the proposed model is limited in that sense.

---

> ### Author Response · Authors · 2022-11-17
> **Response to Reviewer y6fQ**
>
> We thank the reviewer for the detailed feedback. We respond to the concerns raised by the reviewer.
>
> > **R-y1 (nonlinearity):** The non-linear component is cleared for simplicity, but it is not clear whether the activation is not necessary or it can play a role to improve the performance more.
>
> Thank you for the suggestion. We performed an additional experiment to replace the logistic regression in SlenderGNN into a multilayer perceptron (MLP) with nonlinear activation functions. The result shows that nonlinearity mostly hurts the performance of SlenderGNN. Please see Table 4 and the RQ4 in Section 6 for the result and discussion.
>
> > **R-y2 (not a GNN):** In particular, the proposed method is just a logistic regression model that uses neighbor information, connectivity information, and node attributes as features. Eventually, the proposed model solves the iid logistic regression regardless of any latent space. Apparently, the proposed model does not seem to be a GNN. Authors need to modify the definition or the explanation of modeling, or regard the proposed model as the linear model using graph features. Robustness and interpretability can be achieved typically through such simple models.
>
> We think this is a philosophical question on “what defines a GNN.” Most GNNs stack a series of nonlinear layers to learn low-dimensional embeddings of nodes, but linear GNNs that learn no such embeddings have also been very popular these days. For example, SGC (Wu et al., 2019) is a linear GNN, as well as logistic regression with propagated features. It was cited more than 1,400 times and has been used as one of the most popular baselines of GNN research. Based on these works, we aimed to fortify the connection between linear GNNs and (more traditional) logistic regression, which are much related but not discussed actively together in literature. Nevertheless, we are open to changing our method’s name if necessary.
>
> If one thinks that nonlinearity is an essential component for GNNs, we’d like to mention that our design of the propagator function can still be accompanied with nonlinearity, as our new result in Table 4 suggests. However, it is not guaranteed to consistently bring better performance.
>
> > **R-y3 (scalability):** Many GNN models rely on sampling methods, while the proposed method is based on more matrix operation. It would be great if the proposed method can work with some sampled data or scale enough.
>
> In fact, scalability is one of the most notable advantages of our SlenderGNN, which runs instantly even in large graphs where most GNN methods run out of memory. Moreover, as its features are precomputed before training and need no updates, it is not definitive that it can benefit from sampling for speedup. We included four large datasets where many of existing models fail due to the out-of-memory error. Please see Figure 1b and Table 3 for the result.

---

> ### Author Response · Authors · 2022-11-29
> **Looking forward to your response**
>
> Dear reviewer,
>
> Thank you again for your thoughtful review. Based on your suggestions, we clarified the strengths of our SlenderGNN and included new results to support them. We would love to hear your thoughts on our response. Please let us know if there is anything else we can do to address your comments.

---

### Official Review · Reviewer_VYo8 · 2022-10-30

**Confidence:** 3
**Correctness:** 3
**Technical Novelty And Significance:** 2
**Empirical Novelty And Significance:** 3
**Recommendation:** 6

**Clarity, Quality, Novelty And Reproducibility:**

Clarity: In all, the flow is good to me, where I can easily follow the whole story quickly. However, it seems that there is a lack of the explanation about the linear property of SlenderGNN.

Quality: The definitions are consistent with existing literature. The claims are correct imo, including the concrete linearization of existing GNNs elaborated on in the supplementary materials. The sanity checking is designed to accurately reflect the desired properties of a GNN. The experiments are conducted on a comprehensive collection of node classification tasks with various kinds of SOTA GNNs.

Novelty: The discussions of linearization is novel to me. However, the concrete design of the proposed SlenderGNN is straightforward and sacrifices several advantages of existing SOTA spectral GNNs, which raises my concerns about its novelty.

Reproducibility: It is convincing as the authors have provided the repository.

**Strength And Weaknesses:**

Strength:
1. This paper is well written, where all the definitions, declarations, and tables are self-contained.
2. The discussion of linearizing GNN is comprehensive and valuable for the community, which unifies many SOTA GNN works.
3. The performance of SlenderGNN is amazing, especially considering that outperforming those SOTA GNNs on such a variety of node classification tasks is very difficulty in nowadays.

Weaknesses:
1. The SlenderGNN is not a sophisticated instance among the linearization framework, that is to say, it has obvious limitations such as a restricted receptive field. In another word, it is not that novel to me.
2. As a linear GNN, the transformation $P$ could be interpreted from the perspective of feature engineering. Thus, it might be necessary to include several works in that line as baselines (e.g., graph kernel methods) in the experiments.

**Summary Of The Paper:**

This paper reviews the desiderata of GNN and proposes SlenderGNN, a simple linear GNN, to meet the needs. The proposed SlenderGNN comes from the linearization framework, which can resemble various GNN designs. Towards these desiderata, the authors present several sanity check, and the proposed SlenderGNN pass all of them. Moreover, SlenderGNN shows superiority over various GNNs on several real-world node-classification tasks.

**Summary Of The Review:**

I appreciate the discussion of what an ideal GNN is supposed to do, which makes the design of SlenderGNN well-motivated. Although the discussed linearization is somewhat straightforward, it well resembles various kinds of GNNs. I think it is helpful for the community to view existing GNNs from such a perspective. However, I have concerns about the novelty of SlenderGNN, which seems to be a trivial linearization. The consistent advantages achieved by SlenderGNN makes me eventually give a relatively positive score.

---

> ### Author Response · Authors · 2022-11-17
> **Response to Reviewer VYo8**
>
> We thank the reviewer for the positive and detailed comments. We respond to the concerns raised by the reviewer.
>
> > **R-V1 (receptive field and novelty):** The SlenderGNN is not a sophisticated instance among the linearization framework, that is to say, it has obvious limitations such as a restricted receptive field. In another word, it is not that novel to me. The concrete design of the proposed SlenderGNN is straightforward and sacrifices several advantages of existing SOTA spectral GNNs, which raises my concerns about its novelty.
>
> **Receptive field:** We’d like to emphasize that the limited receptive field is our intended way of making SlenderGNN simple but effective. We performed an additional experiment to change its receptive field, and the result shows that the two-hop aggregation works very well in most cases. Please see the RQ4 in Section 6, Table 9, and Appendix F for the result and discussion.
>
> **Novelty:** We believe that designing a simple, effective model is important to the community, and it can be more difficult than designing complex models tailored to each task. SlenderGNN is carefully designed based on the insights from our GNNLin framework and proposed sanity checks, which are also our important contributions. Throughout the updated paper, we added in-depth discussion and more experiments to show how this simplicity is beneficial in various ways, i.e., accuracy, robustness, efficiency, and interpretability. Please see our new results and updated discussion in Section 6.
>
> > **R-V2 (graph kernel methods):** As a linear GNN, the transformation P could be interpreted from the perspective of feature engineering. Thus, it might be necessary to include several works in that line as baselines (e.g., graph kernel methods) in the experiments.
>
> Thank you for the suggestion. We included graph kernel methods as related works in Section 2 and implemented three popular methods as new baselines. Our new results on both sanity checks (in Table 2) and real-world experiments (in Table 3) show that SlenderGNN outperforms them consistently due to more effective design of the propagator function which can handle multiple different scenarios of graphs.
>
> > **R-V3 (linear property):** It seems that there is a lack of the explanation about the linear property of SlenderGNN.
>
> Many advantages of SlenderGNN, especially its interpretability and scalability, come from its linearity. SlenderGNN is interpretable, since the learned weights directly connect the features and predictions, and is scalable, since it does not require backpropagation or the computation of hidden layers. We added more discussion on its properties throughout the paper. Specifically, please check our results and discussion of the RQ2 and RQ3 in Section 6.

---

> > ### Comment · Reviewer_VYo8 · 2022-11-26
> > **Discussion**
> >
> > Thanks for your detailed response! The restriction in receptive field would be a problem in some scenarios, although I also know that those popular datasets may not need a quite deep GNN. Thanks for the supplemented baselines also. In a word, I appreciate the simplicity and would keep the score unchanged.

---

### Official Review · Reviewer_f9sS · 2022-10-30

**Confidence:** 3
**Correctness:** 4
**Technical Novelty And Significance:** 3
**Empirical Novelty And Significance:** 2
**Recommendation:** 6

**Clarity, Quality, Novelty And Reproducibility:**

This paper may benefit from a discussion on why replacing nonlinearities with linear activation functions led to the improvement in accuracy. How the loss in model expressiveness due to removing nonlinearities can lead to the improved performance?
Can the authors mention any improvements upon their presented method that they can address in the future work?


**Strength And Weaknesses:**

This paper follows a strict logical path. The authors first describe their linearization framework. They proceed to describe lessons learned when applying linearization to previous GNN variants. The chief problem they listed have to do with the lack of explainability of GNN models with many layers.
I was particularly impressed with the sanity checks proposed in this paper for evaluating robustness of the proposed method and for comparing it with previous methods.


**Summary Of The Paper:**

The main objectives of this paper are two-fold.
First, the authors introduced a linearization framework for transforming various GNNs into a linearized form. Removing non-linearities should improve interpretability.
Second, the authors applied what they have learned about different GNNs (pain points and factors that distinguish different GNN variants) in designing of their own GNN-variant which is optimized for interpretability, robust to various labeling scenarios in graphs and has fewer parameters.
The authors presented their results together with sanity checks where they identified three groups of graph scenarios composed of combination of Edge types, Labels types and Feature types.


**Summary Of The Review:**

Overall, I found this paper to be well-written and well-motivated. I believe this paper addresses an important problem and provides an innovating solution.

---

> ### Author Response · Authors · 2022-11-17
> **Response to Reviewer f9sS**
>
> We thank the reviewer for the positive and detailed comments. We respond to the concerns raised by the reviewer.
>
> > **R-f1 (nonlinearity):** This paper may benefit from a discussion on why replacing nonlinearities with linear activation functions led to the improvement in accuracy.
>
> Thank you for the suggestion. In fact, the main reason for the success of SlenderGNN is the effective design of its components (i.e., Equation 3), which addresses several pain points of existing models, rather than its linearity, though it gives advantages in various aspects, e.g., interpretability and scalability. To better understand the effect of linearity, we performed an additional experiment to replace the logistic regression in SlenderGNN into a multilayer perceptron (MLP) with nonlinear activation functions. Please see Table 4 and the RQ4 in Section 6 for the result and discussion.
>
> > **R-f2 (expressiveness):** How the loss in model expressiveness due to removing nonlinearities can lead to the improved performance?
>
> We believe that the general performance of a model is not proportional to its expressiveness, especially in node classification, where it is not easy for a complex model to learn an accurate propagation rule from limited data. High expressiveness is likely to cause overfitting and increase the difficulty of hyperparameter tuning, as observed in the success of recent linear GNNs. We aimed at designing an effective model with carefully chosen components for better generalization. Please see our updated Section 4.1 and 4.2 on how SlenderGNN addresses the limitations of existing models.
>
> > **R-f3 (future work):** Can the authors mention any improvements upon their presented method that they can address in the future work?
>
> One possible improvement is to apply our method to different types of graphs such as heterogeneous graphs with multiple node and edge types, or temporal graphs with timestamps. In such cases, we need to design a new propagator function that treats neighboring nodes differently based on such attributes.

---

> ### Author Response · Authors · 2022-11-29
> **Looking forward to your response**
>
> Dear reviewer,
>
> Thank you again for your thoughtful review. Based on your suggestions, we clarified the strengths of our SlenderGNN and included new results to support them. We would love to hear your thoughts on our response. Please let us know if there is anything else we can do to address your comments.

---

### Author Response · Authors · 2022-11-17
**Response to all reviewers**

We thank all reviewers for the detailed and constructive feedback. We updated our paper with additional experiments to address the concerns of reviewers. We colored the updated parts in the paper as pink.

We added a mark, e.g., “R-f1” or “R-V2” at the right side of the paper to indicate which part corresponds to which feedback of a reviewer. We used the first letter of each reviewer’s anonymous name: f9sS, VYo8, y6fQ, and DLLT. Please see our detailed responses to see how we numbered each feedback.

We summarize the changes in the paper before presenting individual responses. First, we performed various new experiments and included them in the paper:
- New ablation studies
  - We added two new ablation studies on SlenderGNN
    - Comparing SlenderGNN with its nonlinear variants (Table 4)
    - Changing the receptive field (i.e., K) of SlenderGNN (Table 9)
  - We moved the original ablation study to Appendix F
    - Verifying the effectiveness of each idea of SlenderGNN (Table 10)
- New large datasets
  - We added 4 large graphs to all experiments (Table 3, 4, 9, and 10)
    - 2 homophily graphs: Ogbn-arXiv and Ogbn-Products
    - 2 heterophily graphs: Twitch and Pokec
- New experiment
  - We performed a scalability experiment on the large datasets (Figure 1b)
- New baselines
  - We added 3 graph kernel methods to all experiments (Table 2 and 3)
    - Reg. Kernel, Diff. Kernel, and RW Kernel

Second, we updated the presentation of our manuscript for better clarity:
- Abstract
  - We emphasized on the ‘careful simplicity’ principle that we follow in this work.
- Introduction (Section 1)
  - We added our philosophy on why we aim to design a simple, effective model as our proposed approach, and why we expect it to perform well on real-world data. We also strengthened the motivation behind the design of SlenderGNN.
- Figure 1 (Section 1)
  - We moved the interpretability part to Section 6, and replaced the old result on model size with new scalability results on the new large datasets.
- Related works (Section 2)
  - We added graph kernel methods as related works, and moved the definition of LR from Section 3 to here.
- SlenderGNN (Section 4)
  - We moved the “pain points of existing models” from Section 3 to here. This aims to clarify that the accuracy improvement of SlenderGNN comes from addressing the pain points summarized from the existing models, instead of the linearization. We also strengthened the justification of our design decisions (D1 - D4) with a more direct connection with the pain points.
- Experiments (Section 6)
  - We added research questions that we aim to answer through experiments. We added detailed discussion on the results, including the new ones on scalability and ablation studies.
- Appendix A
  - We added a table of frequently used symbols.
- Appendix E
  - We present the detailed settings of our experiments here, including the exact definitions of the graph kernel methods we added.
- Appendix F
  - We present the results of ablation studies, including the old one that shows the effectiveness of each idea of SlenderGNN.

---

### Decision · Program_Chairs · 2023-01-20

**Decision:**

Reject

**Justification For Why Not Higher Score:**

This is a borderline paper. With reviewer-AC discussion, I am convinced that the paper does not meet the bar.

**Justification For Why Not Lower Score:**

N/A

**Metareview: Summary, Strengths And Weaknesses:**

This paper proposes a linearization framework for transforming various GNNs into a linearized form, and proposes a simple logistic regression model, SlenderGNN, for node classification.  The strong points include analyzing some pain points of existing GNNs, proposing a simple model combining interpretability, robustness to various labeling scenarios, and a small number of parameters. However, the weaknesses below outweigh the strengths:

Firstly, the proposed model boils down to a logistic regression over the concatenation of 4 handcrafted features, which is of limited technical depth. I believe in some specific scenarios it can outperform advanced GNNs but in general GNNs with multi-layer graph convolutions and nonlinearity will be much more powerful, both theoretically and empirically as shown in many other papers. The results in Table 3 are under a extremely sparse setting where 2.5%/2.5%/95% train/val/test split is used. This is different from the normal 60%/20%/20% setting widely adopted in the GPRGNN and other papers focusing on heterophily, but is never discussed. This raises the question whether the proposed model only works better on the sparse label setting. The setting of the two OGB datasets seem also different from the official one. If the setting is different from standard, it definitely needs a note.

Secondly, the paper claims it can automatically filter out those useless node/structure features, which explains why in the synthetic datasets the model works very well. However, those advanced GNN models can also handle such settings, given you simply use all-one node features as input (when node features are useless), or use fully-connected/node-isolated graph as input (when graph structure is useless). Both of these settings are easy to determine in practice, and being able to automatically adjust or adapt to these (rare) settings is not a significant contribution.

Thirdly, the paper emphasizes its interpretability, yet the only experiment presented in Figure 2 is an illustration of the logistic regression weights which not suprisingly, aligns with the data setting. It is expected to see more than this, e.g., an interpretability evaluation over human-annotated datasets like other interpretability papers do. Those papers can provide explanations for which part of the graph or which feature dimensions determine the prediction.

Overall, I feel this paper still lacks a real solid contribution to the field. It is known that GNNs will fail if graph structure does not make sense, or node features are just noise. However, modern spectral GNNs could universally approximate any filter functions on a graph, thus theoretically can handle all different settings. The paper need to compare with those advanced spectral GNNs (besides GPRGNN) too.

**Summary Of Ac-Reviewer Meeting:**

Unfortuantely, only Reviewer DLLT and I could make to the meeting. During the meeting, I held the same opinion as Reviewer DLLT that this paper may not reach the acceptance bar. Our concerns are summarized in the following.

Firstly, the proposed model is simply a logistic regression over 4 types of handcrafted features, which has limited novelty. I believe in some specific scenarios it can outperform advanced GNNs but in general GNNs with nonlinearity and graph convolution will be much more powerful, both theoretically and empirically as shown in many other papers. The results in Table 3 are under a extremely sparse setting where 2.5%/2.5%/95% train/val/test split is used. This is different from the normal 60%/20%/20% setting widely adopted in the GPRGNN and follow-up papers, but is never discussed.

Secondly, the paper claims it can automatically filter out those useless node/structure features, which explains why in the synthetic datasets the model works very well. However, those advanced GNN models can also handle such settings, given you simply use all-one node features as input (when node features are useless), or use fully-connected/node-isolated graph as input (when graph structure is useless). Both of these settings are easy to determine in practice, and being able to automatically adjust or adapt to these (rare) settings is not a significant contribution.

Thirdly, the paper emphasizes its interpretability, yet the only experiment presented in Figure 2 is an illustration of the logistic regression weights which not suprisingly, aligns with the data setting. It is expected to see more than this, e.g., an interpretability evaluation over human-annotated datasets like many GNN interpretability papers.

Overall, I feel this paper still lacks some real contribution to the field. It is known that GNNs will fail if graph structure does not make sense, or node features are just noise. However, modern spectral GNNs could universally approximate any filter functions on a graph, thus theoretically can handle all different settings. The paper does not compare with those advanced spectral GNNs (except GPRGNN) either.